# Fair Regression with Wasserstein Barycenters

**Evgenii Chzhen**[*]
LMO, Université Paris-Saclay
CNRS, Inria

**Christophe Denis**
LAMA, Université Gustave Eiffel
MIA-Paris, AgroParisTech
INRAE, Université Paris-Saclay

**Mohamed Hebiri**
LAMA, Université Gustave Eiffel
CREST, ENSAE, IP Paris

**Luca Oneto**
DIBRIS, University of Genoa

**Massimiliano Pontil**
Istituto Italiano di Tecnologia
University College London

## Abstract

We study the problem of learning a real-valued function that satisfies the Demographic Parity constraint. It demands the distribution of the predicted output to be independent of the sensitive attribute. We consider the case that the sensitive attribute is available for prediction. We establish a connection between fair regression and optimal transport theory, based on which we derive a close form expression for the optimal fair predictor. Specifically, we show that the distribution of this optimum is the Wasserstein barycenter of the distributions induced by the standard regression function on the sensitive groups. This result offers an intuitive interpretation of the optimal fair prediction and suggests a simple post-processing algorithm to achieve fairness. We establish risk and distribution-free fairness guarantees for this procedure. Numerical experiments indicate that our method is very effective in learning fair models, with a relative increase in error rate that is inferior to the relative gain in fairness.

## 1 Introduction

A central goal of algorithmic fairness is to ensure that sensitive information does not "unfairly" influence the outcomes of learning algorithms. For example, if we wish to predict the salary of an applicant or the grade of a university student, we would like the algorithm to not unfairly use additional sensitive information such as gender or race. Since today's real-life datasets often contain discriminatory bias, standard machine learning methods behave unfairly. Therefore, a substantial effort is being devoted in the field to designing methods that satisfy "fairness" requirements, while still optimizing prediction performance, see for example [5, 10, 13, 16, 18, 21, 23, 25, 26, 28, 32, 45–47, 49] and references therein.

In this paper we study the problem of learning a real-valued regression function which among those complying with the Demographic Parity fairness constraint, minimizes the mean squared error. Demographic Parity requires the probability distribution of the predicted output to be independent of the sensitive attribute and has been used extensively in the literature, both in the context of classification and regression [1, 12, 20, 24, 34]. In this paper we consider the case that the sensitive

---

[*]evgenii.chzhen@universite-paris-saclay.fr

attribute is available for prediction. Our principal result is to show that the distribution of the optimal fair predictor is the solution of a Wasserstein barycenter problem between the distributions induced by the unfair regression function on the sensitive groups. This result builds a bridge between fair regression and optimal transport, [see *e.g.,* 38, 41].

We illustrate our result with an example. Assume that $X$ represents a candidate's skills, $S$ is a binary attribute representing two groups of the population (*e.g.,* majority or minority), and $Y$ is the current market salary. Let $f^*(x, s) = \mathbb{E}[Y|X{=}x, S{=}s]$ be the regression function, that is, the optimal prediction of the salary currently in the market for candidate $(x, s)$. Due to bias present in the underlying data distribution, the induced distribution of market salary predicted by $f^*$ varies across the two groups. We show that the optimal fair prediction $g^*$ transforms the regression function $f^*$ as

$$g^*(x, s) = p_s f^*(x, s) + (1 - p_s) t^*(x, s) \ ,$$

where $p_s$ is the frequency of group $s$ and the correction $t^*(x, s)$ is determined so that the *ranking* of $f^*(x, s)$ relative to the distribution of $X|S = s$ for group $s$ (*e.g.,* minority) is the same as the ranking of $t^*(x, s)$ relative to the distribution of the group $s' \neq s$ (*e.g.,* majority). We elaborate on this example after Theorem 2.3 and in Figure 1. The above expression of the optimal fair predictor naturally suggests a simple post-processing estimation procedure, where we first estimate $f^*$ and then transform it to get an estimator of $g^*$. Importantly, the transformation step involves only unlabeled data since it requires estimation of cumulative distribution functions.

**Contributions and organization.** In summary we make the following contributions. First, in Section 2 we derive the expression for the optimal function which minimizes the squared risk under Demographic Parity constraints (Theorem 2.3). This result establishes a connection between fair regression and the problem of Wasserstein barycenters, which allows to develop an intuitive interpretation of the optimal fair predictor. Second, based on the above result, in Section 3 we propose a post-processing procedure that can be applied on top of any off-the-shelf estimator for the regression function, in order to transform it into a fair one. Third, in Section 4 we show that this post-processing procedure yields a fair prediction independently from the base estimator and the underlying distribution (Proposition 4.1). Moreover, finite sample risk guarantees are derived under additional assumptions on the data distribution provided that the base estimator is accurate (Theorem 4.4). Finally, Section 5 presents a numerical comparison of the proposed method *w.r.t.* the state-of-the-art.

**Related work.** Unlike the case of fair classification, fair regression has received limited attention to date; we are only aware of few works on this topic that are supported by learning bounds or consistency results for the proposed estimator [1, 34]. Connections between algorithmic fairness and Optimal Transport, and in particular the problem of Wasserstein barycenters, has been studied in [12, 20, 24, 43] but mainly in the context of classification. These works are distinct from ours, in that they do not show the link between the optimal fair regression function and Wasserstein barycenters. Moreover, learning bounds are not addressed therein. Our distribution-free fairness guarantees share similarities with contributions on prediction sets [30, 31] and conformal prediction literature [42, 48] as they also rely on results on rank statistics. Meanwhile, the risk guarantee that we derive, combines deviation results on Wasserstein distances in one dimension [7] with peeling ideas developed in [3], and classical theory of rank statistics [40].

**Notation.** For any positive integer $N \in \mathbb{N}$ we denote by $[N]$ the set $\{1, \ldots, N\}$. For $a, b \in \mathbb{R}$ we denote by $a \wedge b$ (*resp.* $a \vee b$) the minimum (*resp.* the maximum) between $a$ and $b$. For two positive real sequences $a_n, b_n$ we write $a_n \lesssim b_n$ to indicate that there exists a constant $c$ such that $a_n \leq c b_n$ for all $n$. For a finite set $\mathcal{S}$ we denote by $|\mathcal{S}|$ its cardinality. The symbols $\mathbf{E}$ and $\mathbf{P}$ stand for generic expectation and probability. For any univariate probability measure $\mu$, we denote by $F_\mu$ its Cumulative Distribution Function (CDF) and by $Q_\mu : [0, 1] \to \mathbb{R}$ its quantile function (*a.k.a.* generalized inverse of $F_\mu$) defined for all $t \in (0, 1]$ as $Q_\mu(t) = \inf \{y \in \mathbb{R} \ : \ F_\mu(y) {\geq} t\}$ with $Q_\mu(0) = Q_\mu(0+)$. For a measurable set $A \subset \mathbb{R}$ we denote by $U(A)$ the uniform distribution on $A$.

## 2 The problem

In this section we introduce the fair regression problem and present our derivation for the optimal fair regression function alongside its connection to Wasserstein barycenter problem. We consider the general regression model

$$Y = f^*(X, S) + \xi \ , \tag{1}$$

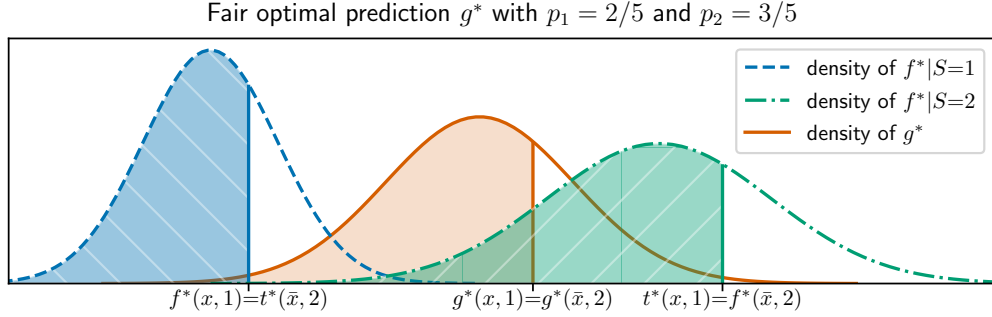

Figure 1: For a new point $(x, 1)$, the value $t^*(x, 1)$ is chosen such that the shaded `Green Area` (//) $=$ $\mathbb{P}(f^*(X, S) \leq t^*(x, 1)|S = 2)$ equals to the shaded `Blue Area` (\\) $= \mathbb{P}(f^*(X, S) \leq f^*(x, 1)|S = 1)$. The final prediction $g^*(x, 1)$ is a convex combination of $f^*(x, 1)$ and $t^*(x, 1)$. The same is done for $(\bar{x}, 2)$.

where $\xi \in \mathbb{R}$ is a centered random variable, $(X, S) \sim \mathbb{P}_{X,S}$ on $\mathbb{R}^d \times \mathcal{S}$, with $|\mathcal{S}| < \infty$, and $f^* : \mathbb{R}^d \times \mathcal{S} \to \mathbb{R}$ is the regression function minimizing the squared risk. Let $\mathbb{P}$ be the joint distribution of $(X, S, Y)$. For any prediction rule $f : \mathbb{R}^d \times \mathcal{S} \to \mathbb{R}$, we denote by $\nu_{f|s}$ the distribution of $f(X, S)|S = s$, that is, the Cumulative Distribution Function (CDF) of $\nu_{f|s}$ is given by

$$F_{\nu_{f|s}}(t) = \mathbb{P}(f(X, S) \leq t|S = s) \ , \tag{2}$$

to shorten the notation we will write $F_{f|s}$ and $Q_{f|s}$ instead of $F_{\nu_{f|s}}$ and $Q_{\nu_{f|s}}$ respectively.

**Definition 2.1** (Wasserstein-2 distance). *Let $\mu$ and $\nu$ be two univariate probability measures. The squared Wasserstein-2 distance between $\mu$ and $\nu$ is defined as*

$$\mathcal{W}_2^2(\mu, \nu) = \inf_{\gamma \in \Gamma_{\mu,\nu}} \int |x - y|^2 \, d\gamma(x, y) \ ,$$

*where $\Gamma_{\mu,\nu}$ is the set of distributions (couplings) on $\mathbb{R} \times \mathbb{R}$ such that for all $\gamma \in \Gamma_{\mu,\nu}$ and all measurable sets $A, B \subset \mathbb{R}$ it holds that $\gamma(A \times \mathbb{R}) = \mu(A)$ and $\gamma(\mathbb{R} \times B) = \nu(B)$.*

In this work we use the following definition of (strong) Demographic Parity, which was previously used in the context of regression by [1, 12, 24].

**Definition 2.2** (Demographic Parity). *A prediction (possibly randomized) $g : \mathbb{R}^d \times \mathcal{S} \to \mathbb{R}$ is fair if, for every $s, s' \in \mathcal{S}$*

$$\sup_{t \in \mathbb{R}} \left| \mathbf{P}(g(X, S) \leq t|S = s) - \mathbf{P}(g(X, S) \leq t|S = s') \right| = 0 \ .$$

Demographic Parity requires the Kolmogorov-Smirnov distance between $\nu_{g|s}$ and $\nu_{g|s'}$ to vanish for all $s, s'$. Thus, if $g$ is fair, $\nu_{g|s}$ does not depend on $s$ and to simplify the notation we will write $\nu_g$.

Recall the model in Eq. (1). Since the noise has zero mean, the minimization of $\mathbb{E}(Y - g(X, S))^2$ over $g$ is equivalent to the minimization of $\mathbb{E}(f^*(X, S) - g(X, S))^2$ over $g$. The next theorem shows that the optimal fair predictor for an input $(x, s)$ is obtained by a nonlinear transformation of the vector $(f^*(x, s))_{s=1}^{|\mathcal{S}|}$ that is linked to a Wasserstein barycenter problem [2].

**Theorem 2.3** (Characterization of fair optimal prediction). *Assume, for each $s \in \mathcal{S}$, that the univariate measure $\nu_{f^*|s}$ has a density and let $p_s = \mathbb{P}(S = s)$. Then,*

$$\min_{g \text{ is fair}} \mathbb{E}(f^*(X, S) - g(X, S))^2 = \min_{\nu} \sum_{s \in \mathcal{S}} p_s \mathcal{W}_2^2(\nu_{f^*|s}, \nu) \ .$$

*Moreover, if $g^*$ and $\nu^*$ solve the l.h.s. and the r.h.s. problems respectively, then $\nu^* = \nu_{g^*}$ and*

$$g^*(x, s) = \left( \sum_{s' \in \mathcal{S}} p_{s'} Q_{f^*|s'} \right) \circ F_{f^*|s} \left( f^*(x, s) \right) \ . \tag{3}$$

The proof of Theorem 2.3 relies on the classical characterization of optimal coupling in one dimension (stated in Theorem A.1 in the appendix) of the Wasserstein-2 distance. We show that a minimizer $g^*$ of the $L_2$-risk can be used to construct $\nu^*$ and vice-versa, given $\nu^*$, we leverage a well-known expression for one dimensional Wasserstein barycenter (see *e.g.,* [2, Section 6.1] and Lemma A.2 in the appendix) and construct $g^*$.

**The case of binary protected attribute.** Let us unpack Eq. (3) in the case that $\mathcal{S} = \{1, 2\}$, assuming *w.l.o.g.* that $p_2 \geq p_1$. Theorem 2.3 states that the fair optimal prediction $g^*$ is defined for all individuals $x \in \mathbb{R}^d$ in the first group as

$$g^*(x, 1) = p_1 f^*(x, 1) + p_2 t^*(x, 1), \text{ with } t^*(x, 1) = \inf \left\{ t \in \mathbb{R} : F_{f^*|2}(t) \geq F_{f^*|1}(f^*(x, 1)) \right\} ,$$

and likewise for the second group. This form of the optimal fair predictor, and more generally Eq. (3), allows us to understand the decision made by $g^*$ at individual level. If we interpret $(x, s)$ as the candidate's CV and candidate's group respectively, and $f^*(x, s)$ as the current market salary (which might be discriminatory), then the fair optimal salary $g^*(x, s)$ is a convex combination of the market salary $f^*(x, s)$ and the adjusted salary $t^*(x, s)$, which is computed as follows. ebgin If say $s=1$, we first compute the fraction of individuals from the first group whose market salary is at most $f^*(x, 1)$, that is, we compute $\mathbb{P}(f^*(X, S) \leq f^*(x, 1)|S=1)$. Then, we find a candidate $\bar{x}$ in group 2, such that the fraction of individuals from the second group whose market salary is at most $f^*(\bar{x}, 2)$ is the same, that is, $\bar{x}$ is chosen to satisfy $\mathbb{P}(f^*(X, S) \leq f^*(\bar{x}, 2)|S=2) = \mathbb{P}(f^*(X, S) \leq f^*(x, 1)|S=1)$. Finally, the market salary of $\bar{x}$ is exactly the adjustment for $x$, that is, $t^*(x, 1) = f^*(\bar{x}, 2)$. This idea is illustrated in Figure 1 and leads to the following philosophy: if candidates $(x, 1)$ and $(\bar{x}, 2)$ share the same group-wise market salary ranking, then they should receive the same salary determined by the fair prediction $g^*(x, 1) = g^*(\bar{x}, 2) = p_1 f^*(x, 1) + p_2 f^*(\bar{x}, 2)$. At last, note that Eq. (3) allows to understand the (potential) amount of extra money that we need to pay in order to satisfy fairness. While the unfair decision made with $f^*$ costs $f^*(x, 1) + f^*(\bar{x}, 2)$ for the salary of $(x, 1)$ and $(\bar{x}, 2)$, the fair decision $g^*$ costs $2(p_1 f^*(x, 1) + p_2 f^*(\bar{x}, 2))$. Thus, the extra (signed) salary that we pay is $\Delta = (p_2 - p_1)(f^*(\bar{x}, 2) - f^*(\bar{x}, 1))$. Since, $p_2 \geq p_1$, $\Delta$ will be positive whenever the candidate $\bar{x}$ from the majority group gets higher salary according to $f^*$, and negative otherwise. We believe that the expression Eq. (3) could be the starting point for further more applied work on algorithmic fairness.

## 3    General form of the estimator

In this section we propose an estimator of the optimal fair predictor $g^*$ that relies on the plug-in principle. The expression (3) of $g^*$ suggests that we only need estimators for the regression function $f^*$, the proportions $p_s$, as well as the CDF $F_{f^*|s}$ and the quantile function $Q_{f^*|s}$, for all $s \in \mathcal{S}$. While the estimation of $f^*$ needs labeled data, all the other quantities rely only on $\mathbb{P}_S$, $\mathbb{P}_{X|S}$ and $f^*$, therefore *unlabeled* data with an estimator of $f^*$ suffices. Thus, given a base estimator of $f^*$, our post-processing algorithm will require only unlabeled data.

For each $s \in \mathcal{S}$ let $\mathcal{U}^s = \{X_i^s\}_{i=1}^{N_s} \overset{\text{i.i.d.}}{\sim} \mathbb{P}_{X|S=s}$ be a group-wise unlabeled sample. In the following for simplicity we assume that $N_s$ are *even* for all $s \in \mathcal{S}^2$. Let $\mathcal{I}_0^s, \mathcal{I}_1^s \subset [N_s]$ be any fixed partition of $[N_s]$ such that $|\mathcal{I}_0^s| = |\mathcal{I}_1^s| = N_s/2$ and $\mathcal{I}_0^s \cup \mathcal{I}_1^s = [N_s]$. For each $j \in \{0, 1\}$ we let $\mathcal{U}_j^s = \{X_i^s \in \mathcal{U}^s : i \in \mathcal{I}_j^s\}$ be the restriction of $\mathcal{U}^s$ to $\mathcal{I}_j^s$. We use $\mathcal{U}_0^s$ to estimate $Q_{f|s}$ and $\mathcal{U}_1^s$ to estimate $F_{f|s}$. For each $f : \mathbb{R}^d \times \mathcal{S} \to \mathbb{R}$ and each $s \in \mathcal{S}$, we estimate $\nu_{f|s}$ by

$$\hat{\nu}_{f|s}^0 = \frac{1}{|\mathcal{I}_0^s|} \sum_{i \in \mathcal{I}_0^s} \delta\big(f(X_i^s, s) + \varepsilon_{is} - \cdot\big) \quad \text{and} \quad \hat{\nu}_{f|s}^1 = \frac{1}{|\mathcal{I}_1^s|} \sum_{i \in \mathcal{I}_1^s} \delta\big(f(X_i^s, s) + \varepsilon_{is} - \cdot\big) , \quad (4)$$

where $\delta$ is the Dirac measure and all $\varepsilon_{is} \overset{\text{i.i.d.}}{\sim} U([-\sigma, \sigma])$, for some positive $\sigma$ set by the user. Using the estimators in Eq. (4), we define for all $f : \mathbb{R}^d \times \mathcal{S} \to \mathbb{R}$ estimators of $Q_{f|s}$ and of $F_{f|s}$ as

$$\hat{Q}_{f|s} \equiv Q_{\hat{\nu}_{f|s}^0} \quad \text{and} \quad \hat{F}_{f|s} \equiv F_{\hat{\nu}_{f|s}^1} . \quad (5)$$

That is, $\hat{F}_{f|s}$ and $\hat{Q}_{f|s}$ are the empirical CDF and empirical quantiles of $(f(X, S) + \varepsilon)|S=s$ based on $\{f(X_i^s, s) + \varepsilon_{is}\}_{i \in \mathcal{I}_1^s}$ and $\{f(X_i^s, s) + \varepsilon_{is}\}_{i \in \mathcal{I}_0^s}$ respectively. The noise $\varepsilon_{is}$ serves as a smoothing random variable, since for all $s \in \mathcal{S}$ and $i \in [N_s]$ the random variables $f(X_i^s, s) + \varepsilon_{is}$ are

**Algorithm 1:** Procedure to evaluate estimator in Eq. (6)

---

**Input:** new point: $(x, s)$; base estimator $\hat{f}$; unlabeled data $\mathcal{U}^1, \ldots, \mathcal{U}^{|\mathcal{S}|}$;
jitter parameter $\sigma$; empirical frequencies $\hat{p}_1, \ldots, \hat{p}_{|S|}$
**Output:** fair prediction $\hat{g}(x, s)$ for the point $(x, s)$

**for** $s' \in \mathcal{S}$ **do**                                                       `// data structure for Eq.(4)`
$\quad \mathcal{U}_0^{s'}, \mathcal{U}_1^{s'} \leftarrow \texttt{split\_in\_two}(\mathcal{U}^{s'})$           `// split unlabeled data into two equal parts`
$\quad \texttt{ar}_0^{s'} \leftarrow \left\{\hat{f}(X, s') + U([-\sigma, \sigma])\right\}_{X \in \mathcal{U}_0^{s'}}, \quad \texttt{ar}_1^{s'} \leftarrow \left\{\hat{f}(X, s') + U([-\sigma, \sigma])\right\}_{X \in \mathcal{U}_1^{s'}}$
$\quad \texttt{ar}_0^{s'} \leftarrow \texttt{sort}\left(\texttt{ar}_0^{s'}\right), \quad \texttt{ar}_1^{s'} \leftarrow \texttt{sort}\left(\texttt{ar}_1^{s'}\right)$           `// for fast evaluation of Eq.(5)`
**end**

$k_s \leftarrow \texttt{position}\left(\hat{f}(x, s) + U([-\sigma, \sigma]), \; \texttt{ar}_1^s\right)$           `// evaluate` $\hat{F}_{\hat{f}|s}\left(\hat{f}(x, s) + \varepsilon\right)$ `in Eq.(6)`
$\hat{g}(x, s) \leftarrow \sum_{s' \in \mathcal{S}} \hat{p}_{s'} \times \texttt{ar}_0^{s'}\left[\lceil N_{s'} k_s / N_s \rceil\right]$           `// evaluation of Eq.(6)`

---

i.i.d. continuous for any $\mathbb{P}$ and $f$. In contrast, $f(X_i^s, s)$ might have atoms resulting in a non-zero probability to observe ties in $\{f(X_i^s, s)\}_{i \in \mathcal{I}_j^s}$. This step is also known as *jittering*, often used for data visualization [11] for tie-breaking. It plays a crucial role in the distribution-free fairness guarantees that we derive in Proposition 4.1; see the discussion thereafter.

Finally, let $\mathcal{A} = \{S_i\}_{i=1}^N \overset{\text{i.i.d.}}{\sim} \mathbb{P}_S$ and for each $s \in \mathcal{S}$ let $\hat{p}_s$ be the empirical frequency of $S = s$ evaluated on $\mathcal{A}$. Given a base estimator $\hat{f}$ of $f^*$ constructed from $n$ labeled samples $\mathcal{L} = \{(X_i, S_i, Y_i)\}_{i=1}^n \overset{\text{i.i.d.}}{\sim} \mathbb{P}$, we define the final estimator $\hat{g}$ of $g^*$ for all $(x, s) \in \mathbb{R}^d \times \mathcal{S}$ mimicking Eq. (3) as

$$\hat{g}(x, s) = \left(\sum_{s' \in \mathcal{S}} \hat{p}_{s'} \hat{Q}_{\hat{f}|s'}\right) \circ \hat{F}_{\hat{f}|s}\left(\hat{f}(x, s) + \varepsilon\right) \quad , \tag{6}$$

where $\varepsilon \sim U([-\sigma, \sigma])$ is assumed to be independent from every other random variables.

**Remark 3.1.** *In practice one should use a very small value for $\sigma$ (e.g., $\sigma = 10^{-5}$), which does not alter the statistical quality of the base estimator $\hat{f}$ as indicated in Theorem 4.4.*

A pseudo-code implementation of $\hat{g}$ in Eq. (6) is reported in Algorithm 1. It requires two primitives: $\texttt{sort(ar)}$ sorts the array $\texttt{ar}$ in an increasing order; $\texttt{position}(a, \texttt{ar})$ which outputs the index $k$ such that the insertion of $a$ into $k$'th position in $\texttt{ar}$ preserves ordering (*i.e.,* $\texttt{ar}[k-1] \leq a < \texttt{ar}[k]$). Algorithm 1 consists of two for parts: in the for-loop we perform a preprocessing which takes $\sum_{s \in \mathcal{S}} O(N_s \log N_s)$ time[3] since it involves sorting; then, the evaluation of $\hat{g}$ on a new point $(x, s)$ is performed in $(\max_{s \in \mathcal{S}} \log N_s)$ time since it involves an element search in a sorted array. Note that the for-loop of Algorithm 1 needs to be performed only once as this step is shared for any new $(x, s)$.

## 4 Statistical analysis

In this section we provide a statistical analysis of the proposed algorithm. We first present in Proposition 4.1 distribution-free finite sample fairness guarantees for post-processing of *any* base learner with unlabeled data and then we show in Theorem 4.4 that if the base estimator $\hat{f}$ is a good proxy for $f^*$, then under mild assumptions on the distribution $\mathbb{P}$, the processed estimator $\hat{g}$ in Eq. (6) is a good estimator of $g^*$ in Eq. (3).

**Distribution free post-processing fairness guarantees.** We derive two distribution-free results in Proposition 4.1, the first in Eq. (7) shows that the fairness definition is satisfied as long as we take the expectation over the data inside the supremum in Definition 2.2, while the second one in Eq. (8) bounds the expected violation of Definition 2.2.

**Proposition 4.1** (Fairness guarantees). *For any joint distribution $\mathbb{P}$ of $(X, S, Y)$, any base estimator $\hat{f}$ constructed on labeled data, and for all $s, s' \in \mathcal{S}$, the estimator $\hat{g}$ defined in Eq. (6) satisfies*

$$\sup_{t \in \mathbb{R}} |\mathbf{P}(\hat{g}(X, S) \leq t | S=s) - \mathbf{P}(\hat{g}(X, S) \leq t | S=s')| \leq 2 \left(N_s \wedge N_{s'} + 2\right)^{-1} \mathbf{1}_{\{N_s \neq N_{s'}\}} \quad (7)$$

$$\mathbf{E} \sup_{t \in \mathbb{R}} |\mathbf{P}(\hat{g}(X, S) \leq t | S=s, \mathcal{D}) - \mathbf{P}(\hat{g}(X, S) \leq t | S=s', \mathcal{D})| \leq 6 \left(N_s \wedge N_{s'} + 1\right)^{-1/2} \,. \quad (8)$$

*where $\mathcal{D} = \mathcal{L} \cup \mathcal{A} \cup_{s \in \mathcal{S}} \mathcal{U}^s$ is the union of all available datasets.*

Let us point out that this result does not require any assumption on the distribution $\mathbb{P}$ as well as on the base estimator $\hat{f}$. This is achieved thanks to the jittering step in the definition of $\hat{g}$ in Eq. (6), which artificially introduces continuity. Continuity allows us to use results from the theory of rank statistics of exchangeable random variables to derive Eq. (7) as well as the classical inverse transform (see *e.g.,* [40, Sections 13 and 21]) combined with the Dvoretzky-Kiefer-Wolfowitz inequality [33] to derive Eq. (8). Since basic results on rank statistics and inverse transform are distribution-free as long as the underlying random variable is continuous, the guarantees in Eqs. (7)–(8) are also distribution-free and can be applied on top of *any* base estimator $\hat{f}$.

The bound in Eq. (7) might be surprising to the reader. Yet, let us emphasize that this bound holds because the expectation *w.r.t.* the data distribution is taken inside the supremum (since $\mathbf{P}$ stands for the joint distribution of *all* random variables involved in $\hat{g}(X, S)$). Similar proof techniques are also used in randomization inference via permutations [19, 22], conformal prediction [30, 42], knockoff estimation [4] to name a few. However, unlike the aforementioned contributions, the problem of fairness requires a non-trivial adaptation of these techniques. In contrast, Eq. (8) might be more appealing to the machine learning community as it controls the expected (over data) violation of the fairness constraint with standard parametric rate.

**Estimation guarantee with accurate base estimator.** In order to prove non-asymptotic risk bounds we require the following assumption on the distribution $\mathbb{P}$ of $(X, S, Y) \in \mathbb{R}^d \times \mathcal{S} \times \mathbb{R}$.

**Assumption 4.2.** *For each $s \in \mathcal{S}$ the univariate measure $\nu_{f^*|s}$ admits a density $q_s$, which is lower bounded by $\underline{\lambda}_s > 0$ and upper-bounded by $\overline{\lambda}_s \geq \underline{\lambda}_s$.*

Although the lower bound on the density assumption is rather strong and might potentially be violated in practice, it is still reasonable in certain situations. We believe that it can be replaced by the assumption that $f^*(X, S)$ conditionally on $S=s$ for all $s \in \mathcal{S}$ admits $2+\epsilon$ moments. We do not explore this relaxation in our work as it significantly complicates the proof of Theorem 4.4. At the same time, our empirical study suggests that the lower bound on the density is not intrinsic to the problem, since the estimator exhibits a good performance across various scenarios. In contrast, the milder assumption that the density is upper bounded is crucial for our proof and seems to be necessary.

Apart from the assumption on the density of $\nu_{f^*|s}$, the actual rate of estimation depends on the quality of the base estimator $\hat{f}$.

**Assumption 4.3.** *There exists a positive constant $c$ independent from $n$, $N$, $N_1, \ldots, N_{|\mathcal{S}|}$, and a positive sequence $b_n : \mathbb{N} \to \mathbb{R}_+$ such that*

$$\mathbf{E}|f^*(X, S) - \hat{f}(X, S)| \leq c b_n^{-1/2} \,.$$

The bound in Assumption 4.3 is stated in a very generic form, it only requires that the base estimator $\hat{f}$ estimates $f^*$ in $\mathbb{L}_1$-norm at the rate $b_n^{-1/2}$. We refer to [3, 15, 29, 30, 39] for various examples of estimators, it includes local polynomial estimators, k-nearest neighbours, and linear regression, to name just a few.

Under these assumptions we can prove the following finite-sample estimation bound.

**Theorem 4.4** (Estimation guarantee). *Let Assumptions 4.2 and 4.3 be satisfied, and set $\sigma \lesssim \min_{s \in \mathcal{S}} N_s^{-1/2} \wedge b_n^{-1/2}$, then the estimator $\hat{g}$ defined in Eq. (6) satisfies*

$$\mathbf{E} |g^*(X, S) - \hat{g}(X, S)| \lesssim b_n^{-1/2} \bigvee \left(\sum_{s \in \mathcal{S}} p_s N_s^{-1/2}\right) \bigvee \sqrt{\frac{|\mathcal{S}|}{N}} \,,$$

*where the leading constant depends only on $\underline{\lambda}_s, \overline{\lambda}_s, c$ from Assumptions 4.2 and 4.3.*

The proof of this result combines expected deviation of empirical measure from the real measure in terms of Wasserstein distance on real line [7] with the already mentioned rank statistics and classical peeling argument of [3].

The first term of the derived bound corresponds to the estimation error of $f^*$ by $\hat{f}$, the second term is the price to pay for not knowing conditional distributions $X|S = s$ while the last term correspond to the price of unknown marginal probabilities of each protected attribute. Notice that if $N_s = p_s N$, which corresponds to the standard i.i.d. sampling from $\mathbb{P}_{X,S}$ of unlabeled data, the second and the third term are of the same order. Moreover, if $N$ is sufficiently large, which in most scenarios[4] is *w.l.o.g.*, then the rate is dominated by $b_n^{-1/2}$. Notice that one can find a collection of joint distributions $\mathbb{P}$, such that $f^*$ satisfies demographic parity. Hence, if $b_n^{-1/2}$ is the minimax optimal estimation rate of $f^*$, then it is also optimal for $g^* \equiv f^*$.

## 5 Empirical study

In this section, we present numerical experiments[5] with the proposed fair regression estimator defined in Section 3. In all experiments, we collect statistics on the test set $\mathcal{T} = \{(X_i, S_i, Y_i)\}_{i=1}^{n_{\text{test}}}$. The empirical mean squared error (MSE) is defined as

$$\text{MSE}\,(g) = \frac{1}{n_{\text{test}}} \sum_{(X,S,Y)\in\mathcal{T}} (Y - g(X,S))^2 \ .$$

We also measure the violation of fairness constraint imposed by Definition 2.2 via the empirical Kolmogorov-Smirnov (KS) distance,

$$\text{KS}\,(g) = \max_{s,s'\in\mathcal{S}} \sup_{t\in\mathbb{R}} \left| \frac{1}{|\mathcal{T}^s|} \sum_{(X,S,Y)\in\mathcal{T}^s} \mathbf{1}_{\{g(X,S)\leq t\}} - \frac{1}{|\mathcal{T}^{s'}|} \sum_{(X,S,Y)\in\mathcal{T}^{s'}} \mathbf{1}_{\{g(X,S)\leq t\}} \right| \ ,$$

where for all $s\in\mathcal{S}$ we define the set $\mathcal{T}^s = \{(X,S,Y) \in \mathcal{T} : S=s\}$. For all datasets we split the data in two parts (70% train and 30% test), this procedure is repeated 30 times, and we report the average performance on the test set alongside its standard deviation. We employ the 2-steps 10-fold CV procedure considered by [17] to select the best hyperparameters with the training set. In the first step, we shortlist all the hyperparameters with MSE close to the best one (in our case, the hyperparameters which lead to 10% larger MSE *w.r.t.* the best MSE). Then, from this list, we select the hyperparameters with the lowest KS.

**Methods.** We compare our method (see Section 3) to different fair regression approaches for both linear and non-linear regression. In the case of linear models we consider the following methods: Linear RLS plus [6] (RLS+Berk), Linear RLS plus [34] (RLS+Oneto), and Linear RLS plus Our Method (RLS+Ours), where RLS is the abbreviation of Regularized Least Squares.
In the case of non-linear models we compare to the following methods. i) For Kernel RLS (KRLS): KRLS plus [34] (KRLS+Oneto), KRLS plus [35] (KRLS+Perez), KRLS plus Our Method (KRLS+Ours); ii) For Random Forests (RF): RF plus [36] (RF+Raff), RF plus [1][6] (RF+Agar), and RF plus Our Method (RF+Ours).
The hyperparameters of the methods are set as follows. For RLS we set the regularization hyperparameters $\lambda \in 10^{\{-4.5,-3.5,\cdots,3\}}$ and for KRLS we set $\lambda \in 10^{\{-4.5,-3.5,\cdots,3\}}$ and $\gamma \in 10^{\{-4.5,-3.5,\cdots,3\}}$. Finally, for RF we set to 1000 the number of trees and for the number of features to select during the tree creation we search in $\{d^{1/4}, d^{1/2}, d^{3/4}\}$.

**Datasets.** In order to analyze the performance of our methods and test it against the state-of-the-art alternatives, we consider five benchmark datasets, CRIME, LAW, NLSY, STUD, and UNIV, which are briefly described below:
*Communities&Crime (CRIME)* contains socio-economic, law enforcement, and crime data about communities in the US [37] with 1994 examples. The task is to predict the number of violent crimes per $10^5$ population (normalized to $[0, 1]$) with race as the protected attribute. Following [9], we made

| Method | CRIME MSE | CRIME KS | LAW MSE | LAW KS | NLSY MSE | NLSY KS | STUD MSE | STUD KS | UNIV MSE | UNIV KS |
|---|---|---|---|---|---|---|---|---|---|---|
| RLS | .033±.003 | .55±.06 | .107±.010 | .15±.02 | .153±.016 | .73±.07 | 4.77±.49 | .50±.05 | 2.24±.22 | .14±.01 |
| RLS+Berk | .037±.004 | .16±.02 | .121±.013 | .10±.01 | .189±.019 | .49±.05 | 5.28±.57 | .32±.03 | 2.43±.23 | .05±.01 |
| RLS+Oneto | .037±.004 | .14±.01 | .112±.012 | .07±.01 | .156±.016 | .50±.05 | 5.02±.54 | .23±.02 | 2.44±.26 | .05±.01 |
| RLS+Ours | .041±.004 | .12±.01 | .141±.014 | .02±.01 | .203±.019 | .09±.01 | 5.62±.52 | .04±.01 | 2.98±.32 | .02±.01 |
| KRLS | .024±.003 | .52±.05 | .040±.004 | .09±.01 | .061±.006 | .58±.06 | 3.85±.36 | .47±.05 | 1.43±.15 | .10±.01 |
| KRLS+Oneto | .028±.003 | .19±.02 | .046±.004 | .05±.01 | .066±.007 | .06±.01 | 4.07±.39 | .18±.02 | 1.46±.13 | .04±.01 |
| KRLS+Perez | .033±.003 | .25±.02 | .048±.005 | .04±.01 | .065±.007 | .08±.01 | 3.97±.38 | .14±.02 | 1.50±.15 | .06±.01 |
| KRLS+Ours | .034±.004 | .09±.01 | .056±.005 | .01±.01 | .081±.008 | .03±.01 | 4.46±.43 | .03±.01 | 1.71±.16 | .02±.01 |
| RF | .020±.002 | .45±.04 | .046±.005 | .11±.01 | .055±.006 | .55±.06 | 3.59±.39 | .45±.05 | 1.31±.13 | .10±.01 |
| RF+Raff | .030±.003 | .21±.02 | .058±.006 | .06±.01 | .066±.006 | .08±.01 | 4.28±.40 | .09±.01 | 1.38±.12 | .02±.01 |
| RF+Agar | .029±.003 | .13±.01 | .050±.005 | .04±.01 | .065±.006 | .07±.01 | 3.87±.41 | .07±.01 | 1.40±.13 | .02±.01 |
| RF+Ours | .033±.003 | .08±.01 | .064±.006 | .02±.01 | .070±.007 | .03±.01 | 4.18±.38 | .02±.01 | 1.49±.14 | .01±.01 |

Table 1: Results for all the datasets and all the methods concerning MSE and KS.

a binary sensitive attribute $s$ as to the percentage of black population, which yielded 970 instances of $s=1$ with a mean crime rate 0.35 and 1024 instances of $s=-1$ with a mean crime rate 0.13.
*Law School (LAW)* refers to the Law School Admissions Councils National Longitudinal Bar Passage Study [44] and has 20649 examples. The task is to predict a students GPA (normalized to $[0, 1]$) with race as the protected attribute (white versus non-white).
*National Longitudinal Survey of Youth (NLSY)* involves survey results by the U.S. Bureau of Labor Statistics that is intended to gather information on the labor market activities and other life events of several groups [8]. Analogously to [27] we model a virtual company's hiring decision assuming that the company does not have access to the applicants' academic scores. We set as target the person's GPA (normalized to $[0, 1]$), with race as sensitive attribute
*Student Performance (STUD)*, approaches 649 students achievement (final grade) in secondary education of two Portuguese schools using 33 attributes [14], with gender as the protected attribute.
*University (UNIV)*[7] is a proprietary and highly sensitive dataset containing all the data about the past and present students enrolled at the University of *Genoa*. In this study we take into consideration students who enrolled, in the academic year 2017-2018. The dataset contains 5000 instances, each one described by 35 attributes (both numeric and categorical) about ethnicity, gender, financial status, and previous school experience. The scope is to predict the average grades at the end of the first semester, with gender as the protected attribute.

**Comparison w.r.t. state-of-the-art.** In Table 1, we present the performance of different methods on various datasets described above. One can notice that LAW and UNIV datasets have a least amount of disciminatory bias (quantified by KS), since the fairness *unaware* methods perform reasonably well in terms of KS. Furthermore, on these two datasets, the difference in performance between all fairness aware methods is less noticeable. In contrast, on CRIME, NLSY, and STUD, fairness unaware methods perform poorly in terms of KS. More importantly, our findings indicate that the proposed method is competitive with state-of-the-art methods and is the most effective in imposing the fairness constraint. In particular, in all except two considered scenarios (CRIME+RLS, CRIME+RF) our method improves fairness by $50\%$ (and up to $80\%$ in some cases) over the closest fairness aware method. In contrast, the accuracy of our method decreases by $1\%$ up to $30\%$ when compared to the most accurate fairness aware method. However, let us emphasize that the relative decrease in accuracy is much smaller than the relative improvement in fairness across the considered scenarios. For example, on NLSY+RLS the most accurate fairness aware method is RLS+Oneto with mean MSE=.156 and mean KS=.50, while RLS+Ours yields mean MSE=.203 and mean KS=.09. That is, compared to RLS+Oneto our method drops about $30\%$ in accuracy, while gains about $82\%$ in fairness. With RF, which is a more powerful estimator, the average drop in accuracy across all datasets compared to RF+Agar is about $12\%$ while the average improvement in fairness is about $53\%$.

# 6 Conclusion and perspectives

In this work we investigated the problem of fair regression with Demographic Parity constraint assuming that the sensitive attribute is available for prediction. We derived a closed form solution for the optimal fair predictor which offers a simple and intuitive interpretation. Relying on this expression, we devised a post-processing procedure, which transforms any base estimator of the regression function into a nearly fair one, independently of the underlying distribution. Moreover, if the base estimator is accurate, our post-processing method yields an accurate estimator of the optimal fair predictor as well. Finally, we conducted an empirical study indicating the effectiveness of our method in imposing fairness in practice. In the future it would be valuable to extend our methodology to the case when we are not allowed to use the sensitive feature as well as to other notions of fairness.

## Broader impact

This work investigates the problem of fair regression with multiple sensitive groups using tools from statistical learning theory and optimal transport theory. Our results lead to an efficient learning algorithm that we show empirically and theoretically to be very effective to impose fairness according to the notion of Demographic Parity. Our approach is directly designed to mitigate potential bias present in the data. Hence, even though the work is primarily theoretical, we anticipate that our results could be used in the future by practitioners in order to specialize our methodology to real-life scenarios involving individuals, and to potentially help making decision which help people with disadvantages or minority groups.

We believe that the most important positive impact of our work is the intuitive interpretation of the optimal fair prediction, which should help to reason as to why a given prediction was made for a given individual. At the same time, this interpretation allows to understand the weaknesses of the notion of Demographic Parity: if $f^*$ does not adequately reflect the group-wise ordering of individuals, the optimal fair prediction $g^*$ might not lead to a fair prediction from individuals' perspective. In other words, returning to the salary example considered above, the notion of Demographic Parity reflects the principle: more qualified individuals get higher salary within their respective groups.

## Acknowledgments and Disclosure of Funding

This work was partially supported Amazon Web Services and SAP SE.

## Footnotes

[2]Since we are ready to sacrifice a factor 2 in our bounds, this assumption is without loss of generality.

[3] It is assumed in this discussion that the time complexity to evaluate $\hat{f}$ is $O(1)$.

[4]One can achieve it by splitting the labeled dataset $\mathcal{L}$ artificially augmenting the unlabeled one, which ensures that $N > n$. In this case if $b_n^{-1/2} = O(n^{-1/2})$, then the first term is always dominant in the derived bound.

[5]The source of our method can be found at `https://github.com/lucaoneto/NIPS2020_Fairness`.

[6]We thank the authors for sharing a prototype of their code.

[7]The data and the research are related to the project DROP@UNIGE of the University of Genoa.

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
