[Supplementary Material]

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

# Supplementary material

The supplementary material is organized as follows. In Appendix A we provide the proof of Theorem 2.3, in Appendix B we provide the proof of Proposition 4.1, and in Appendix C we prove Theorem 4.4. For reader's convenience all the results are repeated in this supplementary material and a short overview of classical results is provided.

## A  Characterization of the optimal

Before providing the proof of Theorem 2.3, let us give a brief overview of classical results in the Optimal transport theory with one dimensional measures; all the results can be found in [39, 43]

**Definition 2.1** (Wasserstein-2 distance). *Let $\mu$ and $\nu$ be two univariate probability measures. The squared Wasserstein-2 distance between $\mu$ and $\nu$ is defined as*

$$\mathcal{W}_2^2(\mu, \nu) = \inf_{\gamma \in \Gamma_{\mu,\nu}} \int |x - y|^2 \, d\gamma(x, y) \ ,$$

*where $\Gamma_{\mu,\nu}$ is the set of distributions (couplings) on $\mathbb{R} \times \mathbb{R}$ such that for all $\gamma \in \Gamma_{\mu,\nu}$ and all measurable sets $A, B \subset \mathbb{R}$ it holds that $\gamma(A \times \mathbb{R}) = \mu(A)$ and $\gamma(\mathbb{R} \times B) = \nu(B)$.*

The coupling $\gamma$ which achieves the infimum in the definition of the Wasserstein-2 distance is called the optimal coupling.

Also let us mention that the Wasserstein-2 distance between two univariate probability measures $\nu, \mu$, defined in Definition 2.1, can be expressed as

$$\mathcal{W}_2^2(\mu, \nu) = \inf_{\gamma} \mathbb{E}_{(Z_\mu, Z_\nu) \sim \gamma}(Z_\nu - Z_\mu)^2 \ ,$$

where $Z_\nu \sim \nu$ and $Z_\mu \sim \mu$ and the infimum is taken over all joint distributions $\gamma$ of $(Z_\nu, Z_\mu)$ which preserve marginals.

The next result establishes that as long as one of the measures in the definition of the Wasserstein-2 distance admits a density, then the optimal coupling in the infimum in Definition 2.1 is deterministic (see *e.g.,* [43, Theorem 2.18] or [39, Theorems 2.5 and 2.9]).

**Theorem A.1.** *Let $\nu, \mu$ be two univariate measures such that $\nu$ has a density and let $X \sim \nu$. Then there exists a mapping $T : \mathbb{R} \to \mathbb{R}$ such*

$$\mathcal{W}_2^2(\mu, \nu) = \mathbb{E}(X - T(X))^2 \ ,$$

*that is $(X, T(X)) \sim \bar{\gamma} \in \Gamma_{\mu,\nu}$ where $\bar{\gamma}$ is an optimal coupling. Moreover, the transport map is given by $T = Q_\mu \circ F_\nu$.*

By the abuse of notation, for an increasing real-valued univariate function $F$ we will use $F^{\leftarrow}$ to denote its generalized inverse. For instance, if $F : \mathbb{R} \to [0, 1]$ is a CDF, then $F^{\leftarrow}$ is the quantile function that was defined in the introduction.

The next result is standard and can be found for instance in [2, Section 6.1] or [39, Section 5.5.5]. It states that for one dimensional Wasserstein barycenter problem, the optimal measure admits a closed form solution.

**Lemma A.2.** *Let $\nu_1, \dots, \nu_{|\mathcal{S}|}$ be $|\mathcal{S}|$ univariate probability measures admitting densities, for all $p_1, \dots, p_{|\mathcal{S}|} \geq 0$ such that $p_1 + \dots + p_{|\mathcal{S}|} = 1$ define*

$$\nu^* \in \arg\min_{\nu} \sum_{s=1}^{|\mathcal{S}|} p_s \mathcal{W}_2^2(\nu_s, \nu) \ .$$

*Then, the cumulative distribution of $\nu^*$ is given by*

$$F_{\nu^*}(\cdot) = \left( \sum_{s=1}^{|\mathcal{S}|} p_s Q_{\nu_s} \right)^{\leftarrow} (\cdot) \ .$$

Theorem A.1 and Lemma A.2 are the two main ingredients that are used in the proof of Theorem 2.3.

**Theorem 2.3** (Characterization of fair optimal prediction). *Assume, for each $s \in \mathcal{S}$, that the univariate measure $\nu_{f^*|s}$ has a density and let $p_s = \mathbb{P}(S = s)$. Then,*

$$\min_{g \text{ is fair}} \mathbb{E}(f^*(X, S) - g(X, S))^2 = \min_{\nu} \sum_{s \in \mathcal{S}} p_s \mathcal{W}_2^2(\nu_{f^*|s}, \nu) \ .$$

*Moreover, if $g^*$ and $\nu^*$ solve the l.h.s. and the r.h.s. problems respectively, then $\nu^* = \nu_{g^*}$ and*

$$g^*(x, s) = \left( \sum_{s' \in \mathcal{S}} p_{s'} Q_{f^*|s'} \right) \circ F_{f^*|s}\left(f^*(x, s)\right) \ . \tag{3}$$

*Proof of Theorem 2.3.* We want to show that

$$\min_{g \text{ is fair}} \mathbb{E}(f^*(X, S) - g(X, S))^2 = \min_{\nu} \sum_{s \in \mathcal{S}} p_s \mathcal{W}_2^2(\nu_{f^*|s}, \nu) \ .$$

Let $\bar{g} : \mathbb{R}^d \times \mathcal{S} \to \mathbb{R}$ be a minimizer of the *l.h.s.* of the above equation and define by $\nu_{\bar{g}}$ the distribution of $\bar{g}$. Since $\nu_{f^*|s}$ admits density, using Theorem A.1 for each $s \in \mathcal{S}$ there exists $T_s = Q_{\nu_{\bar{g}}} \circ F_{f^*|s}$ such that

$$\sum_{s \in \mathcal{S}} p_s \mathcal{W}_2^2(\nu_{f^*|s}, \nu_{\bar{g}}) = \sum_{s \in \mathcal{S}} p_s \int_{\mathbb{R}} (z - T_s(z))^2 \, d\nu_{f^*|s}(z)$$

$$= \sum_{s \in \mathcal{S}} p_s \int_{\mathbb{R}^d} (f^*(x, s) - T_s \circ f^*(x, s))^2 \, d\mathbb{P}_{X|S=s}(x)$$

$$= \sum_{s \in \mathcal{S}} p_s \mathbb{E}\left[ (f^*(X, s) - (T_s \circ f^*)(X, s))^2 \, | S = s \right]$$

$$= \mathbb{E}(f^*(X, S) - \tilde{g}(X, S))^2 \ ,$$

where we defined $\tilde{g}$ for all $(x, s) \in \mathbb{R}^d \times \mathcal{S}$ as

$$\tilde{g}(x, s) = (T_s \circ f^*)(x, s) = \left(Q_{\nu_{\bar{g}}} \circ F_{f^*|s} \circ f^*\right)(x, s) \ .$$

The cumulative distribution of $\tilde{g}$ can be expressed as

$$\mathbb{P}(\tilde{g}(X, S) \leq t) = \sum_{s \in \mathcal{S}} p_s \mathbb{P}_{X|S=s}\left(Q_{\nu_{\bar{g}}} \circ F_{f^*|s} \circ f^*(X, s) \leq t\right)$$

$$= \sum_{s \in \mathcal{S}} p_s \mathbb{P}_{X|S=s}\left(f^*(X, s) \leq Q_{f^*|s} \circ F_{\nu_{\bar{g}}}(t)\right) = F_{\nu_{\bar{g}}}(t) \ ,$$

where the last equality is due to the fact that $\nu_{f^*|s}$ admits a density for all $s \in \mathcal{S}$. The above implies that $\tilde{g}$ is fair, thus on the one hand by optimality of $\bar{g}$ we have

$$\mathbb{E}(f^*(X, S) - \tilde{g}(X, S))^2 \geq \mathbb{E}(f^*(X, S) - \bar{g}(X, S))^2 \ ,$$

on the other hand we have for each $s \in \mathcal{S}$

$$\mathcal{W}_2^2(\nu_{f^*|s}, \nu_{\bar{g}}) \leq \mathbb{E}\left[(f^*(X, S) - \bar{g}(X, S))^2 \, | S = s\right] \ .$$

Thus we showed that

$$\sum_{s \in \mathcal{S}} p_s \mathcal{W}_2^2(\nu_{f^*|s}, \nu_{\bar{g}}) = \min_{g \text{ is fair}} \mathbb{E}(f^*(X, S) - g(X, S))^2 \ . \tag{9}$$

This implies that

$$\min_{\nu} \sum_{s \in \mathcal{S}} p_s \mathcal{W}_2^2(\nu_{f^*|s}, \nu) \leq \min_{g \text{ is fair}} \mathbb{E}(f^*(X, S) - g(X, S))^2 \ . \tag{10}$$

Now we want to show that the opposite inequality also holds. To this end define $\nu^*$ as

$$\nu^* \in \arg\min_{\nu} \sum_{s \in \mathcal{S}} p_s \mathcal{W}_2^2(\nu_{f^*|s}, \nu) \ .$$

Set $T_s^*$ as optimal transport maps from $\nu_{f^*|s}$ to $\nu^*$ of the form $T_s^* = Q_{\nu^*} \circ F_{f^*|s}$ (provided by Theorem A.1 and our assumption on the density of $\nu_{f^*|s}$) and define $g^*$ for all $(x, s) \in \mathbb{R}^d \times \mathcal{S}$ as

$$g^*(x, s) = \left(Q_{\nu^*} \circ F_{f^*|s} \circ f^*\right)(x, s) \ . \tag{11}$$

By the definition of $g^*$ in Eq. (11) and Theorem A.1 we clearly have

$$\min_{\nu} \sum_{s \in \mathcal{S}} p_s \mathcal{W}_2^2(\nu_{f^*|s}, \nu) = \mathbb{E}(f^*(X, S) - g^*(X, S))^2 \ . \tag{12}$$

Moreover since $\nu^*$ is independent from $S$, using similar argument as above one can show that $g^*$ satisfies the Demographic Parity constraint in Definition 2.2 and thus, Eq. (12) yields

$$\min_{\nu} \sum_{s \in \mathcal{S}} p_s \mathcal{W}_2^2(\nu_{f^*|s}, \nu) \geq \min_{g \text{ is fair}} \mathbb{E}(f^*(X, S) - g(X, S))^2 \ . \tag{13}$$

Eqs. (10) and (13) yield the first assertion of the result. Notice that thanks to Eq. (12) we have also shown that

$$\mathbb{E}(f^*(X, S) - g^*(X, S))^2 = \mathbb{E}(f^*(X, S) - \bar{g}(X, S))^2 \ ,$$

and since $g^*$ is fair we can put $\bar{g} = g^*$. Finally, using Lemma A.2 we derive an explicit form of $\nu^*$ and conclude using Eq. (11). $\qquad\square$

## B    Proof of Proposition 4.1

Let us first recall the well-known Dvoretzky–Kiefer–Wolfowitz inequality [34, Corollary 1].

**Theorem B.1** (Dvoretzky–Kiefer–Wolfowitz inequality)**.** *Let $Z_1, \ldots, Z_n$ be i.i.d. real valued random variables with cumulative distribution $F$. Let $\hat{F}$ be the empirical cumulative distribution of $Z_1, \ldots, Z_n$, then*

$$\mathbf{E}\|F - \hat{F}\|_\infty := \mathbf{E}\sup_{t \in \mathbb{R}}|F(t) - \hat{F}(t)| \leq \sqrt{\frac{\pi}{2n}} \ .$$

**Proposition 4.1** (Fairness guarantees)**.** *For any joint distribution $\mathbb{P}$ of $(X, S, Y)$, any base estimator $\hat{f}$ constructed on labeled data, and for all $s, s' \in \mathcal{S}$, the estimator $\hat{g}$ defined in Eq. (6) satisfies*

$$\sup_{t \in \mathbb{R}}|\mathbf{P}(\hat{g}(X, S) \leq t|S{=}s) - \mathbf{P}(\hat{g}(X, S) \leq t|S{=}s')| \leq 2\left(N_s \wedge N_{s'} + 2\right)^{-1} \mathbf{1}_{\{N_s \neq N_{s'}\}} \tag{7}$$

$$\mathbf{E}\sup_{t \in \mathbb{R}}|\mathbf{P}(\hat{g}(X, S) \leq t|S{=}s, \mathcal{D}) - \mathbf{P}(\hat{g}(X, S) \leq t|S{=}s', \mathcal{D})| \leq 6\left(N_s \wedge N_{s'} + 1\right)^{-1/2} \ . \tag{8}$$

*where $\mathcal{D} = \mathcal{L} \cup \mathcal{A} \cup_{s \in \mathcal{S}} \mathcal{U}^s$ is the union of all available datasets.*

*Proof of Proposition 4.1.* The proof of Eq. (7) is based on standard results in the theory of rank statistics (see e.g. [42, Sec. 13]). Meanwhile, the proof of Eq. (8) is built upon the well-known Dvoretzky–Kiefer–Wolfowitz inequality [34, Corollary 1].

Notice that if $X^s \sim \mathbb{P}_{X|S=s}$ and $X^s$ is independent from labeled, unlabeled data, and the noise variables $\varepsilon_{is}, \varepsilon$, then it holds that

$$\mathbf{P}(\hat{g}(X, S) \leq t|S = s) = \mathbf{P}(\hat{g}(X^s, s) \leq t), \quad \forall t \in \mathbb{R} \ .$$

**Proof of Eq. (7):** We have for all $s, s' \in \mathcal{S}$ that

$$\sup_{t \in \mathbb{R}}\left|\mathbf{P}(\hat{g}(X^s, s) \leq t) - \mathbf{P}(\hat{g}(X^{s'}, s') \leq t)\right|$$

$$\leq \sup_{t \in (0,1)}\left|\mathbf{P}\left(\hat{F}_{\hat{f}|s}\left(\hat{f}(X^s, s) + \varepsilon\right) \leq t\right) - \mathbf{P}\left(\hat{F}_{\hat{f}|s'}\left(\hat{f}(X^{s'}, s') + \varepsilon\right) \leq t\right)\right| \ ,$$

where, thanks to the form of $\hat{g}$ in Eq. (6), the inequality follows from the fact that for all $s \in \mathcal{S}$

$$\left(\sum_{\tilde{s} \in \mathcal{S}} \hat{p}_{\tilde{s}}\hat{Q}_{\hat{f}|\tilde{s}}\right) \circ \hat{F}_{\hat{f}|s}\left(\hat{f}(x, s) + \varepsilon\right) \leq t \quad \Leftrightarrow \quad \hat{F}_{\hat{f}|s}\left(\hat{f}(x, s) + \varepsilon\right) \leq \left(\sum_{\tilde{s} \in \mathcal{S}} \hat{p}_{\tilde{s}}\hat{Q}_{\hat{f}|\tilde{s}}\right)^{\leftarrow}(t) \ .$$

Fix some $t \in (0,1)$ and let $k_s(t) \in \{1, \ldots, |\mathcal{I}_1^s|\}$ be such that $\frac{k_s(t)-1}{|\mathcal{I}_1^s|} \leq t < \frac{k_s(t)}{|\mathcal{I}_1^s|}$, then by the definition of $\hat{F}_{\hat{f}|s}(\cdot)$ we have

$$\hat{F}_{\hat{f}|s}\left(\hat{f}(x,s) + \varepsilon\right) \leq t \quad \Leftrightarrow \quad \sum_{i \in \mathcal{I}_1^s} \mathbf{1}_{\left\{\hat{f}(X_i^s,s)+\varepsilon_{is} \leq \hat{f}(x,s)+\varepsilon\right\}} \leq k_s(t) - 1 \ .$$

Notice that the random variables $\{\hat{f}(X^s, s) + \varepsilon\} \cup \{\hat{f}(X_i^s, s) + \varepsilon_{is}\}_{i \in \mathcal{I}_1^s}$ conditionally on labeled data $\mathcal{L}$ are i.i.d. and continuous. Thus, conditionally on $\mathcal{L}$ the random variable $\sum_{i \in \mathcal{I}_1^s} \mathbf{1}_{\left\{\hat{f}(X_i^s,s)+\varepsilon_{is} \leq \hat{f}(X^s,s)+\varepsilon\right\}}$ is distributed uniformly on $\{0, \ldots, |\mathcal{I}_1^s|\}$ (see *e.g.,* [42, Lemma 13.1]), so that

$$\mathbf{P}\left(\hat{F}_{\hat{f}|s}\left(\hat{f}(X^s, s) + \varepsilon\right) \leq t\right) = \frac{k_s(t)}{|\mathcal{I}_1^s| + 1} \ .$$

Repeating the same argument for $s'$ and recalling that $|\mathcal{I}_1^s| = N_s/2$ and $|\mathcal{I}_1^{s'}| = N_{s'}/2$, we get

$$\sup_{t \in \mathbb{R}} \left| \mathbf{P}(\hat{g}(X^s, s) \leq t) - \mathbf{P}(\hat{g}(X^{s'}, s') \leq t) \right| \leq \sup_{t \in (0,1)} \left| \frac{k_s(t)}{N_s/2 + 1} - \frac{k_{s'}(t)}{N_{s'}/2 + 1} \right|$$
$$= 2 \left(N_s \wedge N_{s'} + 2\right)^{-1} \mathbf{1}_{\{N_s \neq N_{s'}\}} \ .$$

**Proof of Eq. (8):** Similarly, as in the proof of Eq. (7) we can write

$$(*) = \sup_{t \in \mathbb{R}} \left| \mathbf{P}(\hat{g}_s(X^s) \leq t | \mathcal{D}) - \mathbf{P}(\hat{g}_{s'}(X^{s'}) \leq t | \mathcal{D}) \right|$$
$$\leq \sup_{t \in (0,1)} \left| \mathbf{P}\left(\hat{F}_{\hat{f}|s}\left(\hat{f}(X^s, s) + \varepsilon\right) \leq t | \mathcal{D}\right) - \mathbf{P}\left(\hat{F}_{\hat{f}|s'}\left(\hat{f}(X^{s'}, s') + \varepsilon\right) \leq t | \mathcal{D}\right) \right| \ .$$

Moreover, thanks to the triangle inequality we have

$$(*) \leq \sup_{t \in (0,1)} \left| \mathbf{P}\left(\hat{F}_{\hat{f}|s}\left(\hat{f}(X^s, s) + \varepsilon\right) \leq t | \mathcal{D}\right) - \mathbf{P}\left(F_{\bar{\nu}_{\hat{f}|s}}\left(\hat{f}(X^s, s) + \varepsilon\right) \leq t | \mathcal{D}\right) \right|$$
$$+ \sup_{t \in (0,1)} \left| \mathbf{P}\left(\hat{F}_{\hat{f}|s'}\left(\hat{f}(X^{s'}, s') + \varepsilon\right) \leq t | \mathcal{D}\right) - \mathbf{P}\left(F_{\bar{\nu}_{\hat{f}|s'}}\left(\hat{f}(X^{s'}, s') + \varepsilon\right) \leq t | \mathcal{D}\right) \right|$$
$$= \sup_{t \in (0,1)} A_s(t) + \sup_{t \in (0,1)} A_{s'}(t) \ , \tag{14}$$

where for all $t \in \mathbb{R}$ and all $s \in \mathcal{S}$ we defined

$$F_{\bar{\nu}_{\hat{f}|s}}(t) = \mathbf{P}\left(\hat{f}(X^s, s) + \varepsilon \leq t | \mathcal{D}\right) \ ,$$

and we used the fact that $\hat{f}(X^s, s) + \varepsilon$ is continuous conditionally on all the available data $\mathcal{D}$, then the random variable $F_{\bar{\nu}_{\hat{f}|s}}\left(\hat{f}(X^s, s) + \varepsilon\right)$ is distributed uniformly on $(0,1)$ (see *e.g.,* [42, Lemma 21.1]), which means that for all $t \in (0,1)$ and all $s, s' \in \mathcal{S}$

$$t = \mathbf{P}\left(F_{\bar{\nu}_{\hat{f}|s}}\left(\hat{f}(X^s, s) + \varepsilon\right) \leq t | \mathcal{D}\right) = \mathbf{P}\left(F_{\bar{\nu}_{\hat{f}|s'}}\left(\hat{f}(X^{s'}, s') + \varepsilon\right) \leq t | \mathcal{D}\right) \ .$$

We bound the first term in Eq. (14) and the bound for the second terms follows the same arguments. Fix some $t \in (0,1)$, then we can write

$$A_s(t) \leq \mathbf{P}\left(\left|F_{\bar{\nu}_{\hat{f}|s}}\left(\hat{f}(X^s, s) + \varepsilon\right) - t\right| \leq \left|F_{\bar{\nu}_{\hat{f}|s}}\left(\hat{f}(X^s, s) + \varepsilon\right) - \hat{F}_{\hat{f}|s}\left(\hat{f}(X^s, s) + \varepsilon\right)\right| \Big| \mathcal{D}\right)$$
$$\leq \mathbf{P}\left(\left|F_{\bar{\nu}_{\hat{f}|s}}\left(\hat{f}(X^s, s) + \varepsilon\right) - t\right| \leq \left\|F_{\bar{\nu}_{\hat{f}|s}} - \hat{F}_{\hat{f}|s}\right\|_\infty \Big| \mathcal{D}\right) \leq 2 \left\|F_{\bar{\nu}_{\hat{f}|s}} - \hat{F}_{\hat{f}|s}\right\|_\infty \ .$$

Taking supremum on both sides and repeating the same argument for $s'$, we get

$$(*) \leq 2\mathbf{E}\left\|F_{\bar{\nu}_{\hat{f}|s}} - \hat{F}_{\hat{f}|s}\right\|_\infty + 2\mathbf{E}\left\|F_{\bar{\nu}_{\hat{f}|s'}} - \hat{F}_{\hat{f}|s'}\right\|_\infty \ ,$$

we conclude applying Dvoretzky–Kiefer–Wolfowitz inequality, recalled in Theorem B.1, conditionally on $\mathcal{L}$. □

# C  Proof of Theorem 4.4

Let us first recall the assumptions that we require in order to prove Theorem 4.4.

**Assumption 4.2.** *For each $s \in \mathcal{S}$ the univariate measure $\nu_{f^*|s}$ admits a density $q_s$, which is lower bounded by $\underline{\lambda}_s > 0$ and upper-bounded by $\overline{\lambda}_s \geq \underline{\lambda}_s$.*

**Assumption 4.3.** *There exists a positive constant $c$ independent from $n$, $N$, $N_1, \ldots, N_{|\mathcal{S}|}$, and a positive sequence $b_n : \mathbb{N} \to \mathbb{R}_+$ such that*

$$\mathbf{E}|f^*(X,S) - \hat{f}(X,S)| \leq c b_n^{-1/2} \ .$$

We also need to define Wasserstein 1 and $\infty$ distances.

**Definition C.1.** *Let $\mu$ and $\nu$ be two univariate probability measures, then Wasserstein 1 and $\infty$ distance between $\mu$ and $\nu$ are defined as*

$$\mathcal{W}_1(\mu,\nu) = \int_0^1 |Q_\mu(t) - Q_\nu(t)| \, dt \quad and \quad \mathcal{W}_\infty(\mu,\nu) = \sup_{t \in [0,1]} |Q_\mu(t) - Q_\nu(t)| \ ,$$

*respectively.*

**Remark C.2.** *The definitions of $\mathcal{W}_1$ and $\mathcal{W}_\infty$ can be stated in terms of couplings as it is done in Definition 2.1. However, for our purposes it is more convenient to use their equivalent formulation stated in Definition C.1. We refer to [7] and in particular to their Theorem 2.10 for further details.*

The final ingredient is [7, Theorem 5.11].

**Theorem C.3.** *Let $Z_1, \ldots, Z_n$ be i.i.d. real valued random variables from some probability measure $\mu$ and let $\hat{\mu}$ be the empirical measure based on $Z_1, \ldots, Z_n$. Assume that $\mu$ admits density which is lower-bounded by some constant $L > 0$, then*

$$\mathbf{E}[\mathcal{W}_\infty(\mu, \hat{\mu})] \leq L^{-1} \sqrt{\frac{2\pi}{n}} \ .$$

**Theorem 4.4** (Estimation guarantee). *Let Assumptions 4.2 and 4.3 be satisfied, and set $\sigma \lesssim \min_{s \in \mathcal{S}} N_s^{-1/2} \wedge b_n^{-1/2}$, then the estimator $\hat{g}$ defined in Eq. (6) satisfies*

$$\mathbf{E} \, |g^*(X,S) - \hat{g}(X,S)| \lesssim b_n^{-1/2} \bigvee \left( \sum_{s \in \mathcal{S}} p_s N_s^{-1/2} \right) \bigvee \sqrt{\frac{|\mathcal{S}|}{N}} \ ,$$

*where the leading constant depends only on $\underline{\lambda}_s, \overline{\lambda}_s, c$ from Assumptions 4.2 and 4.3.*

*Proof of Theorem 4.4.* In the proof $a > 0$ is going to denote an absolute constant independent from the size of data, which can differ from line to line. First of all, define the random variable

$$\Delta(\hat{g}) = \mathbb{E} \, |\hat{g}(X,S) - g^*(X,S)| = \sum_{s \in \mathcal{S}} p_s \mathbb{E} \left[ |\hat{g}(X,s) - g^*(X,s)| \, | S = s \right] \ ,$$

where $\mathbb{E}$ stands for the expectation *w.r.t.* the joint distribution of $(X, S, Y)$. Recall that

$$g^*(x,s) = \sum_{s' \in \mathcal{S}} p_{s'} Q_{f^*|s'} \left( F_{f^*|s} (f^*(x,s)) \right) \text{ and } \hat{g}(x,s) = \sum_{s' \in \mathcal{S}} \hat{p}_{s'} \hat{Q}_{\hat{f}|s'} \left( \hat{F}_{\hat{f}|s} \left( \hat{f}(x,s) + \varepsilon \right) \right) \ .$$

Considering $g^*(x,s)$ first, we can state that

$$\left| g^*(x,s) - \sum_{s' \in \mathcal{S}} \hat{p}_{s'} Q_{f^*|s'} \left( F_{f^*|s} (f^*(x,s)) \right) \right| \leq \sum_{s' \in \mathcal{S}} |p_{s'} - \hat{p}_{s'}| \times \left| Q_{f^*|s'} \circ F_{f^*|s} \circ f^*(x,s) \right| \ .$$

It is clear that if we can find a bound on $|f^*(x,s')|$ which holds for almost all $x$ w.r.t. $\mathbb{P}_{X|S=s'}$, it would imply an upper bound on $|Q_{f^*|s'}(t)|$ for all $t \in [0,1]$. Fix some $a > 0$, then on the one hand for all $s' \in \mathcal{S}$

$$\mathbb{P}(|f^*(X,S)| \leq a | S = s') \leq 1 \ ,$$

on the other hand under Assumption 4.2 we can write for all $s' \in \mathcal{S}$

$$\mathbb{P}(|f^*(X,S)| \leq a|S{=}s') = \int_{|f^*(x,s')|\leq a} \mathbb{P}_{X|S=s'}(dx) = \int_{|t|\leq a} q_{s'}(t)dt \geq \underline{\lambda}_{s'} \int_{|t|\leq a} dt = 2a\underline{\lambda}_{s'} \ ,$$

which implies that $a \leq 1/(2\underline{\lambda}_{s'})$ and therefore $|f^*(x,s')| \leq 1/(2\underline{\lambda}_{s'})$ for almost all $x \in \mathbb{R}^d$ w.r.t. $\mathbb{P}_{X|S=s'}$. Hence, we can write for all $(x,s) \in \mathbb{R}^d \times \mathcal{S}$

$$\left| g^*(x,s) - \sum_{s'\in\mathcal{S}} \hat{p}_{s'} Q_{f^*|s'}\left(F_{f^*|s}\left(f^*(x,s)\right)\right)\right| \leq \frac{1}{2} \sum_{s'\in\mathcal{S}} \underline{\lambda}_{s'}^{-1} |p_{s'} - \hat{p}_{s'}| \ .$$

The above implies that

$$\Delta(\hat{g}) \leq \sum_{s\in\mathcal{S}} p_s \sum_{s'\in\mathcal{S}} \hat{p}_{s'} \mathbb{E}\left[ \left| Q_{f^*|s'}\left(F_{f^*|s}\left(f^*(X,S)\right)\right) - \hat{Q}_{\hat{f}|s'}\left(\hat{F}_{\hat{f}|s}\left(\hat{f}(X,s)+\varepsilon\right)\right)\right| \Big| S=s\right]$$

$$+ \frac{1}{2} \sum_{s\in\mathcal{S}} \underline{\lambda}_s^{-1} |p_s - \hat{p}_s| \ .$$

Taking the total expectation we arrive at

$$\mathbb{E}[\Delta(\hat{g})] \leq \sum_{s,s'\in\mathcal{S}} p_s p_{s'} \mathbb{E}\left[ \left| Q_{f^*|s'}\left(F_{f^*|s}\left(f^*(X,S)\right)\right) - \hat{Q}_{\hat{f}|s'}\left(\hat{F}_{\hat{f}|s}\left(\hat{f}(X,S)+\varepsilon\right)\right)\right| \Big| S=s\right]$$

$$+ \frac{1}{2} \sum_{s\in\mathcal{S}} \underline{\lambda}_s^{-1} \mathbb{E}\,|p_s - \hat{p}_s| \ ,$$

where we used the fact that $\hat{p}_s$ is an unbiased estimator of $p_s$. For all $s \in \mathcal{S}$ let $X^s \sim \mathbb{P}_{X|S=s}$ be independent from everything, for all $s', s \in \mathcal{S}$ set the shorthand notation

$$\mathrm{a}_{ss'} = \mathbb{E}\left| Q_{f^*|s'}\left(F_{f^*|s}\left(f^*(X^s,s)\right)\right) - \hat{Q}_{\hat{f}|s'}\left(\hat{F}_{\hat{f}|s}\left(\hat{f}(X^s,s)+\varepsilon\right)\right)\right| \ .$$

Notice that

$$\mathrm{a}_{ss'} = \mathbb{E}\left[ \left| Q_{f^*|s'}\left(F_{f^*|s}\left(f^*(X,S)\right)\right) - \hat{Q}_{\hat{f}|s'}\left(\hat{F}_{\hat{f}|s}\left(\hat{f}(X,S)+\varepsilon\right)\right)\right| \Big| S=s\right] \ ,$$

and therefore we can write

$$\mathbb{E}\,|\hat{g}(X,S) - g^*(X,S)| = \mathbb{E}[\Delta(\hat{g})] \leq \sum_{s,s'\in\mathcal{S}} p_s p_{s'} \mathrm{a}_{ss'} + \frac{1}{2} \sum_{s\in\mathcal{S}} \underline{\lambda}_s^{-1} \mathbb{E}\,|p_s - \hat{p}_s| \ .$$

Notice that the term $\mathbb{E}\,|p_s - \hat{p}_s| = N^{-1}\mathbb{E}|Np_s - V|$, where $V$ is the binomial random variable with parameters $(N, p_s)$, thus using the Cauchy–Schwarz inequality we can write $\mathbb{E}\,|p_s - \hat{p}_s| \leq N^{-1}\sqrt{\mathrm{Var}(V)} = \sqrt{p_s(1-p_s)/N}$ and the above bound reads as

$$\mathbb{E}\,|\hat{g}(X,S) - g^*(X,S)| \leq \sum_{s,s'\in\mathcal{S}} p_s p_{s'} \mathrm{a}_{ss'} + \frac{1}{2} \sum_{s\in\mathcal{S}} \underline{\lambda}_s^{-1} \sqrt{\frac{p_s(1-p_s)}{N}}$$

$$\leq \sum_{s,s'\in\mathcal{S}} p_s p_{s'} \mathrm{a}_{ss'} + \frac{N^{-1/2}}{2} \max_{s\in\mathcal{S}} \underline{\lambda}_s^{-1} \sum_{s\in\mathcal{S}} \sqrt{p_s(1-p_s)} \ . \qquad (15)$$

It remains to bound $\mathrm{a}_{ss'}$ for each $s, s' \in \mathcal{S}$. Fix some $s, s' \in \mathcal{S}$ (they can be equal), then

$$\mathrm{a}_{ss'} \leq \underbrace{\mathbb{E}\left| \hat{Q}_{f^*|s'}\left(\hat{F}_{\hat{f}|s}\left(\hat{f}(X^s,s)+\varepsilon\right)\right) - \hat{Q}_{\hat{f}|s'}\left(\hat{F}_{\hat{f}|s}\left(\hat{f}(X^s,s)+\varepsilon\right)\right)\right|}_{\mathrm{a}^1_{ss'}}$$

$$+ \underbrace{\mathbb{E}\left| Q_{f^*|s'}\left(\hat{F}_{\hat{f}|s}\left(\hat{f}(X^s,s)+\varepsilon\right)\right) - \hat{Q}_{f^*|s'}\left(\hat{F}_{\hat{f}|s}\left(\hat{f}(X^s,s)+\varepsilon\right)\right)\right|}_{\mathrm{a}^2_{ss'}} \qquad (16)$$

$$+ \underbrace{\mathbb{E}\left| Q_{f^*|s'}\left(F_{f^*|s}\left(f^*(X^s,s)\right)\right) - Q_{f^*|s'}\left(\hat{F}_{\hat{f}|s}\left(\hat{f}(X^s,s)+\varepsilon\right)\right)\right|}_{\mathrm{a}^3_{ss'}} \ .$$

We bound each of the three terms separately.

**First term** ($\mathrm{a}^1_{ss'}$): Notice that $\hat{F}_{\hat{f}|s}\left(\hat{f}(X^s,s)+\varepsilon\right)$ is distributed uniformly on $\{0,1/|\mathcal{I}^s_1|,2/|\mathcal{I}^s_1|,\ldots,1\}$ conditionally on labeled data $\mathcal{L}$ (see *e.g.*, [42, Lemma 13.1]). Thus, we have

$$\mathrm{a}^1_{ss'} = \frac{1}{|\mathcal{I}^s_1|+1} \sum_{j=0}^{|\mathcal{I}^s_1|} \mathbf{E}\left|\hat{Q}_{f^*|s'}\left(\frac{j}{|\mathcal{I}^s_1|}\right) - \hat{Q}_{\hat{f}|s'}\left(\frac{j}{|\mathcal{I}^s_1|}\right)\right| \ . \tag{17}$$

Notice that for all $j\in\{1,\ldots,|\mathcal{I}^s_1|\}$ and all $\alpha\in((j-1)/|\mathcal{I}^s_1|,j/|\mathcal{I}^s_1|]$ it holds that

$$\hat{Q}_{f^*|s'}\left(\frac{j}{|\mathcal{I}^s_1|}\right) = \hat{Q}_{f^*|s'}(\alpha) \ .$$

The above implies that

$$\frac{1}{|\mathcal{I}^s_1|}\hat{Q}_{f^*|s'}\left(\frac{j}{|\mathcal{I}^s_1|}\right) = \int_{j/|\mathcal{I}^s_1|}^{(j+1)/|\mathcal{I}^s_1|} \hat{Q}_{f^*|s'}(\alpha)\,d\alpha \ , \tag{18}$$

and the same argument repeated for $\hat{Q}_{\hat{f}|s'}$ implies that

$$\frac{1}{|\mathcal{I}^s_1|}\hat{Q}_{\hat{f}|s'}\left(\frac{j}{|\mathcal{I}^s_1|}\right) = \int_{j/|\mathcal{I}^s_1|}^{(j+1)/|\mathcal{I}^s_1|} \hat{Q}_{\hat{f}|s'}(\alpha)\,d\alpha \ . \tag{19}$$

Substituting Eqs. (18)-(19) in Eq. (17) and using Definition C.1 we get

$$\mathrm{a}^1_{ss'} \leq 2\mathbf{E}\int_0^1 \left|\hat{Q}_{f^*|s'}(\alpha) - \hat{Q}_{\hat{f}|s'}(\alpha)\right| d\alpha = 2\mathbf{E}\mathcal{W}_1(\hat{\nu}^0_{f^*|s'},\hat{\nu}^0_{\hat{f}|s'}) \ ,$$

where for $j=0$ in Eq. (17) we used the fact that $\frac{1}{|\mathcal{I}^s_1|}\mathbf{E}|\hat{Q}_{f^*|s'}(0) - \hat{Q}_{\hat{f}|s'}(0)| \leq \mathbf{E}\int_0^1|\hat{Q}_{f^*|s'}(\alpha) - \hat{Q}_{\hat{f}|s'}(\alpha)|d\alpha$. Using the coupling definition of the Wasserstein distance and the way we have defined $\hat{\nu}^0_{f|s'}$, we can write

$$\mathcal{W}_1(\hat{\nu}^0_{f^*|s'},\hat{\nu}^0_{\hat{f}|s'}) \leq \frac{1}{|\mathcal{I}^{s'}_0|} \sum_{i\in\mathcal{I}^{s'}_0} \left|f^*(X^{s'}_i,s') + \varepsilon_{is'} - (\hat{f}(X^{s'}_i,s') + \varepsilon_{is'})\right| \ ,$$

almost surely. Since $\{X^{s'}_i\}_{i\in\mathcal{I}^{s'}_0}$ are i.i.d. from $\mathbb{P}_{X|S=s'}$, then conditionally on $\mathcal{L}$ the random variables $\{|f^*(X^{s'}_i,s) - \hat{f}(X^{s'}_i,s')|\}_{i\in\mathcal{I}^{s'}_0}$ are i.i.d. . Furthermore, using Assumption 4.3 we can write

$$\mathrm{a}^1_{ss'} \leq 2\mathbf{E}\mathcal{W}_1(\hat{\nu}^0_{f^*|s'},\hat{\nu}^0_{\hat{f}|s'}) \leq 2\mathbf{E}\left[\left|f^*(X,S) - \hat{f}(X,S)\right|\Big|S=s'\right] \overset{\text{Assumption 4.3}}{\leq} 2\mathtt{A}b_n^{-1/2} \ . \tag{20}$$

**Second term** ($\mathrm{a}^2_{ss'}$): Note that under Assumption 4.2, the Lipschitz constant of $Q_{f^*|s'}$ is upper bounded by $\underline{\lambda}_{s'}^{-1}$. Then, taking supremum and using Definition C.1 we apply Theorem C.3 to get

$$\mathrm{a}^2_{ss'} \leq \mathbf{E}\mathcal{W}_\infty\left(\nu_{f^*|s'},\hat{\nu}^0_{f^*|s'}\right) \leq a\underline{\lambda}_{s'}^{-1}N_{s'}^{-1/2} \ . \tag{21}$$

where $a$ is an absolute positive constant ($a=2\sqrt{2\pi}$ is sufficient).

**Third term** ($\mathrm{a}^3_{ss'}$): We can write, using Assumption 4.2 that

$$\begin{aligned}
\mathrm{a}^3_{ss'} &\leq \underline{\lambda}_{s'}^{-1}\mathbf{E}\left|F_{f^*|s}(f^*(X^s,s)) - \hat{F}_{\hat{f}|s}\left(\hat{f}(X^s,s)+\varepsilon\right)\right| \\
&\leq \underline{\lambda}_{s'}^{-1}\mathbf{E}\left|F_{f^*|s}(f^*(X^s,s)) - F_{\bar{\nu}_{\hat{f}|s}}\left(\hat{f}(X^s,s)+\varepsilon\right)\right| + \underline{\lambda}_{s'}^{-1}\mathbf{E}\|F_{\bar{\nu}_{\hat{f}|s}}(t) - \hat{F}_{\hat{f}|s}(t)\|_\infty \ , \quad (22)
\end{aligned}$$

with $F_{\bar{\nu}_{\hat{f}|s}}$ defined for all $s\in\mathcal{S}$ and all $t\in\mathbb{R}$ as

$$F_{\bar{\nu}_{\hat{f}|s}}(t) = \mathbf{P}\left(\hat{f}(X^s,s)+\varepsilon \leq t\big|\mathcal{L}\right) \ . \tag{23}$$

The second term in Eq. (22) is bounded by $\lesssim 2\underline{\lambda}_{s'}^{-1} N_s^{-1/2}$ thanks to the Dvoretzky–Kiefer–Wolfowitz inequality recalled in Theorem B.1. Thus, it remains to bound the first term in Eq. (22). We introduce the following shorthand notation for the first term in Eq. (22)

$$(*) = \mathbf{E}\left| F_{f^*|s}\left(f^*(X^s, s)\right) - F_{\tilde{\nu}_{\hat{f}|s}}\left(\hat{f}(X^s, s) + \varepsilon\right)\right| \ .$$

Let $\tilde{X}^s \sim \mathbb{P}_{X|S=s}$ and $\tilde{\varepsilon} \sim U[-\sigma, \sigma]$ be independent from $\varepsilon$, $X^s$, $\mathcal{L}$ and each other. Based on this notation we can write

$$(*) = \mathbf{E}\left|\underbrace{\mathbf{P}\left(f^*(\tilde{X}^s, s) - f^*(X^s, s) \leq 0 \big| \varepsilon, X^s, \mathcal{L}\right)}_{H_0} - \underbrace{\mathbf{P}\left(\hat{f}(\tilde{X}^s, s) + \tilde{\varepsilon} \leq \hat{f}(X^s, s) + \varepsilon \big| \varepsilon, X^s, \mathcal{L}\right)}_{H_1}\right| \ . \quad (24)$$

Furthermore, if $\Delta(X^s) = f^*(X^s, s) - \hat{f}(X^s, s)$, $\Delta(\tilde{X}^s) = f^*(\tilde{X}^s, s) - \hat{f}(\tilde{X}^s, s)$, and $\Delta_\varepsilon = \varepsilon - \tilde{\varepsilon}$, then simple algebra yields

$$H_1 = \mathbf{P}\left(f^*(\tilde{X}^s, s) - f^*(X^s, s) \leq \Delta_\varepsilon + \Delta(\tilde{X}^s) - \Delta(X^s) \big| \varepsilon, X^s, \mathcal{L}\right) \ .$$

For all $a, b \in \mathbb{R}$ it holds that $|\mathbf{1}_{\{a \leq 0\}} - \mathbf{1}_{\{a \leq b\}}| \leq \mathbf{1}_{\{0 \wedge b \leq a \leq 0 \vee b\}} \leq \mathbf{1}_{\{-|b| \leq a \leq |b|\}} = \mathbf{1}_{\{|a| \leq |b|\}}$. Applying this fact to Eq. (24) with $a = f^*(\tilde{X}^s, s) - f^*(X^s, s)$ and $b = \Delta_\varepsilon + \Delta(\tilde{X}^s) - \Delta(X^s)$ we get

$$(*) \leq \mathbf{P}\left(\left|f^*(\tilde{X}^s, s) - f^*(X^s, s)\right| \leq |\Delta_\varepsilon| + \left|\Delta(\tilde{X}^s)\right| + \left|\Delta(X^s)\right|\right)$$

$$\leq \mathbf{P}\left(\left|f^*(\tilde{X}^s, s) - f^*(X^s, s)\right| \leq 3|\Delta_\varepsilon|\right) + \mathbf{P}\left(\left|f^*(\tilde{X}^s, s) - f^*(X^s, s)\right| \leq 3\left|\Delta(\tilde{X}^s)\right|\right)$$

$$+ \mathbf{P}\left(\left|f^*(\tilde{X}^s, s) - f^*(X^s, s)\right| \leq 3\left|\Delta(X^s)\right|\right) \ .$$

By definition of $\tilde{X}^s$ the random variables $X^s, \tilde{X}^s$ are exchangeable, hence

$$\mathbf{P}(|f^*(\tilde{X}^s, s) - f^*(X^s, s)| \leq 3|\Delta(\tilde{X}^s)|) = \mathbf{P}(|f^*(\tilde{X}^s, s) - f^*(X^s, s)| \leq 3|\Delta(X^s)|) \ .$$

Furthermore, using the fact that $|\varepsilon - \tilde{\varepsilon}| \leq 2\sigma$ almost surely we get

$$(*) \leq \mathbf{P}\left(\left|f^*(\tilde{X}^s, s) - f^*(X^s, s)\right| \leq 6\sigma\right) + 2\mathbf{P}\left(\left|f^*(\tilde{X}^s, s) - f^*(X^s, s)\right| \leq 3\left|\Delta(X^s)\right|\right) \ . \quad (25)$$

Thanks to Assumption 4.2 we have the following bound on the first term in Eq. (25)

$$\mathbf{P}\left(|f^*(\tilde{X}^s, s) - f^*(X^s, s)| \leq 6\sigma\right) \leq \mathbf{E}\left[\mathbf{P}\left(|f^*(\tilde{X}^s, s) - f^*(X^s, s)| \leq 6\sigma | X^s\right)\right] \leq 12\overline{\lambda}_s \sigma \ .$$

For the second term in Eq. (25), we observe that Assumption 4.2 yields almost surely

$$\mathbf{P}\left(|f^*(\tilde{X}^s, s) - f^*(X^s, s)| \leq 3\left|\Delta(X^s)\right| \big| \mathcal{L}, X^s\right) \leq 6\overline{\lambda}_s \left|\Delta(X^s)\right| \ .$$

Thus, taking the total expectation on both sides of this inequality we get

$$\mathbf{P}\left(|f^*(\tilde{X}^s, s) - f^*(X^s, s)| \leq 3\left|\Delta(X^s)\right|\right) \leq 6\overline{\lambda}_s \mathbf{E}\left|\Delta(X^s)\right| \overset{\text{Assumption 4.3}}{\leq} 6\overline{\lambda}_s \mathbb{A} b_n^{-1/2} \ .$$

Since $\sigma \lesssim b_n^{-1/2}$, then we have demonstrated that $(*) \lesssim \overline{\lambda}_s b_n^{-1/2}$. Substituting this bound into Eq. (22), we derive that

$$\mathrm{a}_{ss'}^3 \lesssim \underline{\lambda}_{s'}^{-1} \overline{\lambda}_s b_n^{-1/2} + \underline{\lambda}_{s'}^{-1} N_s^{-1/2} \ . \quad (26)$$

**Gathering three terms together:** Finally, substituting Eqs. (20), (21), (26) into Eq. (16) we get

$$\mathrm{a}_{ss'} \lesssim b_n^{-1/2} + \underline{\lambda}_{s'}^{-1} \overline{\lambda}_s b_n^{-1/2} + \underline{\lambda}_{s'}^{-1} N_{s'}^{-1/2} + \underline{\lambda}_{s'}^{-1} N_s^{-1/2} \ .$$

Finally, substituting the bound above into Eq. (15) we arrive at

$$\mathbf{E}\left|\hat{g}(X, S) - g^*(X, S)\right| \lesssim b_n^{-1/2} + \left(\sum_{s \in \mathcal{S}} p_s \underline{\lambda}_s^{-1}\right)\left(\sum_{s \in \mathcal{S}} p_s \overline{\lambda}_s\right) b_n^{-1/2}$$

$$+ \sum_{s \in \mathcal{S}} p_s \underline{\lambda}_s^{-1} N_s^{-1/2} + \left(\sum_{s \in \mathcal{S}} p_s \underline{\lambda}_s^{-1}\right)\left(\sum_{s \in \mathcal{S}} p_s N_s^{-1/2}\right)$$

$$+ N^{-1/2} \max_{s \in \mathcal{S}} \underline{\lambda}_s^{-1} \sum_{s \in \mathcal{S}} \sqrt{p_s(1 - p_s)}$$

$$\lesssim b_n^{-1/2} + \sum_{s \in \mathcal{S}} p_s N_s^{-1/2} + \sqrt{|\mathcal{S}|} N^{-1/2} \ ,$$

where in the last inequality we used the fact that

$$\sum_{s \in \mathcal{S}} \sqrt{p_s(1 - p_s)} \leq \sum_{s \in \mathcal{S}} \sqrt{p_s} \leq \sqrt{|\mathcal{S}|} \sqrt{\sum_{s \in \mathcal{S}} p_s} = \sqrt{|\mathcal{S}|} \ .$$

This ends the proof. □

**Remark C.4.** *Notice that the exact constant in front of the rate of convergence in Theorem 4.4 can be recovered following the proof. Furthermore, this proof can be extended to control $L_p$ norm*

$$\left(\mathbf{E}|g^*(X, S) - \hat{g}(X, S)|^p\right)^{1/p} \ ,$$

*for all $p \in [1, \infty)$ (the current proof deals only with $p = 1$). To achieve it one only needs to extend Assumption 4.3 while the rest of the proof follows line-by-line using deviation results on Wasserstein-$p$ distance on the real line [7]. Finally, it is possible to extend this result under the same assumptions to control $\mathbf{E} \|g^* - \hat{g}\|_\infty$, which induces an extra multiplicative polylogarithmic factor in $b_n^{-1/2}$.*

**Remark C.5.** *We remark that Theorem 2.3 can be extended to the case of continuous sensitive attribute $S$. Assume that $S \in \mathbb{R}^L$ and that $S$ has a finite second moment and admits density $p_S : \mathbb{R}^L \to \mathbb{R}_+$ w.r.t. the Lebesgue measure, then formally replacing $\mathbb{P}(S = s)$ in Theorem 2.3 by $p_S(s)$ and the summation sign by the integration sign we would get*

$$g^*(x, s) = \left(\int_{s' \in \mathbb{R}^L} p_S(s') Q_{f^*|s'} ds'\right) \circ F_{f^*|s}(f^*(x, s)) \ .$$

*In order to construct a post-processing data-driven method, we take some base estimator $\hat{f}$ and estimate conditional cumulative distribution function [32] of $\hat{f}(X, S) \mid S = s$ and the marginal distribution $p_S(\cdot)$ [40].*