[Reviews · NeurIPS 2020]

Review 1

Summary and Contributions: The paper derives a closed form solution for the g*, the optimal regressor that satisfies a demographic parity constraint. This suggests a general post-processing approach for transforming the best estimate of f* into the best estimate of g*. Edit post rebuttal: I thank the authors for their response.

Strengths: I found the paper to be clearly written, and the experiments quite thorough. I think it develops the body of knowledge on fair regression (albeit for a very "naive" notion of fairness, demographic parity).

Weaknesses: The empirical study in Section 5 suggests that the proposed approach is better at imposing the DP constraint relative to other approaches, but does so at the cost of accuracy. The authors defend this by claiming that the improvement in fairness is larger in magnitude than the decrease in accuracy. I don't think this is relevant - we know DP comes at a cost to accuracy, so this tradeoff is to be expected. Even the optimal g*, if we could find it, would decrease accuracy at the cost of fairness. So the relevant empirical question here, I think, is whether the proposed approach yields a point that is on the Pareto frontier of accuracy and fairness. I think it would be interesting to explore this empirically. Is there a variant of the algorithm proposed that allows directly controlling for this tradeoff, i.e. rather than trying to approximate the optimal fair predictor g*, approximate the optimal predictor that is no more than \alpha unfair?

Correctness: I am not familiar with optimal transport theory so beyond the general sensibility of the results, I did not verify them. The experimental methodology makes sense.

Clarity: Yes

Relation to Prior Work: Yes

Reproducibility: Yes

Additional Feedback:


Review 2

Summary and Contributions: This paper studies how to design an optimal linear regression model which satisfies demographic parity constraint. The authors achieve this goal by exploring the connection between fair regression and optimal transport theory, Wasserstein barycenter in particular. Such connection leads to a closed-form expression for the optimal fair predictor. This closed-form expression has a very intuitive interpretation and results in a simple post-processing procedure for designing fair predictor. Finally, the authors analyze the statistical properties of the proposed approach and compare their algorithm with several existing baseline approaches.

Strengths: This paper provides a closed-form expression of the optimal fair regression model which leads to a simple post-processing procedure for designing fair predictor. I appreciate the efforts the authors put on analyzing the estimation error of the optimal classifier due to finite sample. In particular, post-processing a (potentially discriminatory but well-calibrated) predictor using unlabeled data has been widely appeared in the classification tasks. However, rarely many works consider the statistical guarantees of the proposed algorithm. For example, what if the pre-trained model is not calibrated; how many data are required to guarantee small estimation error of the designed predictor. I believe this paper provides some useful toolsets and gives promising results towards answering these questions.

Weaknesses: I find the main result (Theorem 2.3) not very surprising because the connection between optimal fair predictor and Wasserstein barycenter has been observed in previous works on classification tasks. Furthermore, Assumption 4.2 seems too strong since it requires the probability density function to be lower bounded by a constant.

Correctness: The results in this paper seem correct.

Clarity: This paper is well-written and the notation is clear although some minor typos exist. I like Figure 1 in particular since it illustrates how the optimal fair predictor is constructed in a very clear and intuitive way. Some typos: Line 114 "begin" The font size in Section 5 seems inconsistent.

Relation to Prior Work: The authors have done a comprehensive literature review on the topic and compared their methods with baseline approaches through experiments on benchmark datasets.

Reproducibility: Yes

Additional Feedback: The philosophical interpretation of the optimal fair predictor is interesting. I wonder if it has any connection with works on causality (e.g., counterfactual fairness). A key technical step in this paper is to add uniform noise (see e.g., Eq(4)). However, the choice of sigma is left to the user. Can the authors provide any insight on how to choose such constant given their analysis in Section 4 (e.g., Theorem 4.4). Also, why uniform noise? For example, it has been observed that adding Gaussian noise to the empirical distribution can significantly improve the convergence rate in high dimension. I understand that here the predictor is in one dimension and adding uniform noise is good enough but can you provide any intuition on why such type of noise is chosen initially? ##### After rebuttal ##### Thanks for sending a well-structured and comprehensive rebuttal!


Review 3

Summary and Contributions: Updated Review: I thank the authors for updating their paper to handle cts attributes, and address the major point in my review. I've updated my review to accept. This paper studies the problem of regression under demographic parity constraints, and makes an interesting connection to the one-dimensional Wasserstein barycenter problem. In particular, they show the problem of computing the optimal fair regressor wrt to l2 loss, is equivalent to the problem of solving the wasserstein barycenter problem in 1 dimension. Since this problem has a closed form solution, this gives a closed form for the optimal fair regressor, as a function of the optimal regressor and the true data distribution. They use this optimal form to construct a new estimator for the optimal fair regressor, based on estimating the quantities involved (quantile functions and CDFs mainly), which can be viewed as a post-processing of optimal unfair classifier. They show that if the optimal unfair classifier is nearly optimal, then under general circumstances, this post-processed classifier will be nearly optimal. Finally they perform an empirical study across a range of datasets, showing that their classifier often substantially increases fairness relative to other fair regression techniques, at the cost of slightly worse accuracy.

Strengths: (1) Derives a new form of the optimal fair regressor for a common notion of fairness (demographic parity) (2) Beyond theory, the derived estimator is useful in some situations

Weaknesses: (1) These results only apply to demographic parity, and not to other notions of statistical fairness like EO - which are more commonly used (2) This method relies crucially on the sensitive feature being discrete, and does not scale well to large number of sensitive features or continuous sensitive attributes - limiting its applicability in many machine learning scenarios (4) This method is compared to methods like RLS + Berk by comparing a single (accuracy, fairness) point, whereas these methods allow us to trace an accuracy fairness curve. How do we know there aren’t points on the curve that pareto dominate this method? (5) The inability to trace out a curve / relax the fairness criterion is a drawback of this method

Correctness: yes

Clarity: - Could be improved; typos throughout. For example, page 4. line 114. - In equation (8) it isn’t clear to me what conditioning on D with respect the probabilities; does this mean the probabilities are calculated in sample (e.g. they are the empirical probabilities?)

Relation to Prior Work: Yes

Reproducibility: Yes

Additional Feedback:


Review 4

Summary and Contributions: The paper considers achieving demographic parity for regression problems. Using optimal transport theory, it shows that the fair regressor can be described in terms of the Wasserstein barycenter of the distributions distributions induced by the standard regression function on the sensitive groups. It provides theoretical guarantees of their algorithm to find such fair regressor and empirically evaluates their algorithm against other state-of-art methods.

Strengths: Because of the closed form expression of the fair regressor, which decomposes into just multiple regressors, the paper provides further intuition of what the demographic parity constraints actually entail. The actual decomposition and how to find the optimal fair regressor through appropriate results from optimal transport literature are quite crisp and nice in my opinion -- there aren't too many complicated details that need to be considered in order to apply the results from optimal transport, yet sufficiently many interesting insights that I appreciated overall. The fact that it just requires post-processing of any off the shelf estimator and that it only requires unlabeled data to be able to find such post-processing makes it quite attractive for practical use as well.

Weaknesses: My biggest worry is that I'm not sure whether this work adds significantly new contributions compared to the previous literature that uses optimal transport theory for fair classification. It seems like it's the modification of the post-processing approach in "Wasserstein Fair Classification" (Jiang et al). I would be happy to increase the score if the authors can highlight some challenges faced in updating approaches from previous work to this regression problem and how these challenges are not trivial. I wish there was a little more discussion about looking at these fairness constrained optimization problems through the lens of optimal transport theory; the paper only considered demographic parity, but maybe a discussion of why it is or is not immediate this approach may work for other fairness notions, such as equalized odds (appropriately 're-defined' for the regression problem). Also, I wish whether it's possible to allow for some slack when considering demographic parity (difference can be at most some epsilon). Then, for empirical evaluation of the algorithms (I'm not sure if other method also allows for such slack though) by trying different epsilon values, one can sketch out the pareto curves (accuracy vs achieved demographic parity) of these algorithms and actually try to see if one algorithm strictly pareto dominates other ones. By doing so, it may be easier to compare the algorithms: for instance, one can fix the achieved demographic parity to be all equal across these algorithms and see which one achieves the best accuracy.

Correctness: Yes. Although I haven't checked all the minute details thoroughly, the claims and methods definitely seem sound and correct.

Clarity: Yes, the paper flows very smoothly with just enough details in the main body and all the necessary details provided in the supplementary. It also interprets some of the theorems (e.g. 'the case of binary protected attribute' at the end of page 3), which actually provides good intuition as to what's going on.

Relation to Prior Work: I haven't spent much time going through the details of the previous contributions thoroughly, but maybe due to the overlapping theme of using optimal transport theory to look at the problem, the techniques from the previous work and this paper seem pretty similar. Besides the fact that this paper considers the regression problem, it would be nice if the authors can describe the non-trivial challenges faced when trying to port over those techniques from previous approaches in classification problem to the regression problem.

Reproducibility: Yes

Additional Feedback: typo on line 114: "ebgin" ----------------------------------------------------------------------------------------------- After the rebuttal: The authors have addressed my concern with respect to its relation to previous work. Although there are some things (allowing for slack in DP constraint) that I would like to have seen addressed more, the paper overall has crisp characterization of the DP optimization that I will raise the score to "7 accept".

[Author Response · NeurIPS 2020]

We thank all reviewers for their valuable comments. Let us first provide a concise recap of our contributions. **i)** We
derive a closed form expression of the minimizer of the squared risk under the Demographic Parity (DP) constraint
(Thm. 2.3). **ii)** We propose an efficient post-processing algorithm (Alg. 1) which can be applied on top of *any*
off-the-shelf estimator of the regression function, requiring *only* unlabeled data. **iii)** Our algorithm achieves strong
finite sample fairness guarantees *without any* assumptions (Prop. 4.1). **iv)** Under additional assumptions, we derive
plug-and-play finite sample risk guarantees (Thm. 4.4). These contributions lead to an intuitive understanding of DP
(ll. 107–128), result in a computationally efficient method (ll. 156–162) which is interpretable and enjoys strong finite
sample statistical guarantees (Section 4). We highlight that contributions **i)** and **iii)** are, up to our knowledge, unique.
We now address specific points raised by the reviewers, which will be included in the final version upon acceptance.
**Optimal Transport (OT) + Fairness (R2, R4):** Let us highlight two key differences between "Wasserstein Fair
Classification" (Jiang et al.) and our work. **1.** While they directly work in the space of distributions and with
transportation maps, we start from the problem of minimizing the risk under the DP constraint over *functions* and
establish a link between the optimization over functions (*l.h.s.* problem in Thm. 2.3) and optimization over distributions
(*r.h.s.* problem). In particular, they do not derive the form of a classifier which minimizes the *misclassification risk* (or
the squared risk) under the DP constraint, a technical challenge that we solved in our paper for regression. **2.** Unlike our
contribution, they neither provide risk guarantees nor they give bounds on the violation of the DP constraint, whereas we
provide finite sample controls of both. Apart from shared spirit of OT, to the best of our knowledge, our contributions
do not follow or generalize any previous work on fairness. On the other hand, our statistical analysis borrows tools from
non-parametric statistics, rank statistics, empirical processes, and statistics in Wasserstein (Wass.) spaces.
**Pareto frontier (R1, R3, R4):** This is an interesting direction of future research. In order to study the Pareto frontier
one needs to study the problem $\min\{\text{risk}(g) : \text{DP}(g) \leq \varepsilon\}$. Note that since DP is defined via the Kolmogorov-Smirnov
(KS) distance it is oblivious to the geometry of the ambient space. In particular, the major technical challenge is to
build a connection between the space of functions with $\mathbb{L}_2$ geometry and the distributions with the KS geometry. Our
analysis establishes this connection for the case of DP $= 0$, by leveraging the fact that Wass. geometry in the space of
distributions is "synchronized" with the squared risk geometry in the space of predictions. Another possible direction is
to find $g^*_\varepsilon$, which minimizes the squared risk under the constraint that the Wass. barycenter objective is bounded by $\varepsilon$.
Yet, this does not directly imply that $g^*_\varepsilon$ is optimal for the problem $\min\{\text{risk}(g) : \text{DP}(g) \leq \varepsilon\}$.
**Other notions of fairness (R3, R4):** It would indeed be interesting to investigate extension of our analysis to other
fairness notions. The main difficulty in such an extension for, e.g., Equalized Odds is due to the conditioning on the
signal $Y$. Notice also that DP is used in several papers, including Jiang et al. discussed above.
**R1. "naive" notion of fairness"** Let us disagree that the notion of DP is naive. Generally group fairness constraints
are trying to reflect a certain independence between the prediction and the sensitive attribute. DP is simply one of
possible independence constraints that is, above all else, widely used in practice.
**R2. "Assumption 4.2"** As stated in the paper (ll. 196–202), we agree with R2 that As. 4.2 might be strong in certain
situations. However a form of this assumption is rather classical in non-parametric statistics (see *e.g.,* "Fast learning
rates for plug-in classifiers" Audibert & Tsybakov; Def. 2.2). In our settings As. 4.2 is mostly technical and can be
further relaxed with much more involved analysis (see ll. 197–198). **"choice of sigma is left to the user."** We care to
point out that Thm. 4.4 gives exact order of $\sigma$ and Rem. 3.1 provides general guidelines. **"how to choose $\sigma$ [...] why
uniform noise?"** R2 raises an important point. Indeed, fairness guarantees (Prop. 4.1) do not require *any* condition on
the noise level $\sigma > 0$, while Thm. 4.4 gives its exact value. This discrepancy is dictated by completely different proof
techniques of Prop. 4.1 and Thm. 4.4 and the fact that DP does not care about the quality of the base estimator. In
particular, for Thm. 4.4 it is important that the noise: *i)* is continuous *ii)* does not deviate far from zero. Meanwhile, in
Prop. 4.1 we only need the continuity of the noise and we do *not* care about its magnitude. Continuous noise allows us
to derive *assumption free* fairness guarantees using tools from rank statistics and empirical processes. One can indeed
use Gaussian noise with small variance. It does not affect Prop. 4.1 and the proof of Thm. 4.4 can be slightly modified.
**R3. " [...] does not scale well to large number of sensitive features".** We disagree with the reviewer. As indicated at
ll. 160–161 our post-processing procedure has worst case *training* complexity $N \log N$ and $\log N$ for *inference* (with
$N$ being the total number of unlabeled data). **"[...] continuous sensitive attribute."** We thank the reviewer for this
comment, it allowed us to extend our results to this case. Informally, it requires to replace $\mathbb{P}(S=s)$ by the density $\varphi(s)$
of random variable $S$ ($\sum$ replaced by $\int$). Consequently, in the method (Eq. (6)) one needs to replace the estimates $\hat{p}_s$
by an estimator $\hat{\varphi}(s)$ of the density $\varphi(s)$ (*e.g.,* KDE). We will include this part in the final version. **"does this mean the
probabilities are calculated in sample?"** Note that all of our bounds are *out of sample*. In Eq.(8) $\mathbf{P}$ stands for the joint
distribution of data $\mathcal{D}$, added noise, and $(X, S)$. Under $\mathbf{P}(\cdot|S=s, \mathcal{D})$, the method $\hat{g}$ is seen as non-random. Randomness
comes only from the point $(X, S)$. **"How do we know there aren't points [...] that Pareto dominate this method".**
It is clear that the predictor $g^*$ that *minimizes* the risk under the constraint that DP$=0$ is Pareto efficient, hence no
other predictor can Pareto dominate $g^*$. Thanks to our finite sample guarantees, we can say that risk$(\hat{g}) \approx$ risk$(g^*)$ and
DP$(\hat{g}) \approx 0$. Thus $\hat{g}$ is nearly Pareto efficient and cannot be dominated by any other method at the population level.
**R4.** Given the above discussion, we hope that the reviewer is convinced that our contributions neither follow trivially
from previous works on OT and fairness, nor can be seen as a straightforward extension to the regression setup.

[Meta-Review · NeurIPS 2020]

The reviewers agreed that the closed-form expression of the optimal fair regression model under demographic parity constraint is a nice contribution. The theoretical and empircal results are solid. The authors should provide a discussion on why they didn't provide a pareto curve in the revision.